

# Mechanisms of clay smear formation in unconsolidated sediments - insights from 3D observations of excavated normal faults

Michael Kettermann[1], Sebastian Thronberens[1,*], Oscar Juarez[2], Janos Lajos Urai[1], Martin Ziegler[2], Sven Asmus[3], and Ulrich Krüger[3]

[1]Structural Geology, Tectonics and Geomechanics Energy and Mineral Resources Group, RWTH Aachen University, Lochnerstraße 4-20, D-52056 Aachen, Germany
[2]Chair of Geotechnical Engineering, RWTH Aachen University, Aachen, Germany
[3]RWE Power AG, Cologne, Germany
[*]Now at Weatherford International plc, Hannover, Germany

*Correspondence to:* M. Kettermann (michael.kettermann@emr.rwth-aachen.de)

**Abstract.** Clay smears in normal faults can form seals for hydrocarbons and groundwater, and their prediction in the subsurface is an important problem in applied and basic geoscience. However, neither their complex 3D structure, nor their processes of formation or destruction are well understood, and outcrop studies to date are mainly 2D. We present a 3D study of an excavated normal fault with clay smear, together with both source layers, in unlithified sand and clay of the Hambach open cast lignite mine in Germany. The faults formed at a depth of 150 m, and have Shale Gouge Ratios between 0.1 and 0.3. The fault zones are layered, with sheared sand, sheared clay and tectonically mixed sand-clay gouge. Thickness of clay smears in two excavated fault zones of 1.8 and 3.8 $m^2$ is approximately log-normal, with values between 5 mm and 5 cm, without holes. The 3D thickness distribution is heterogeneous. We show that clay smears are strongly affected by R- and R'-shears, mostly at the footwall side. These shears can locally cross and offset clay smears, forming holes in the clay smear, while thinning of the clay smear by shearing in the fault core is less important. Thinnest parts of the clay smears are often located close to source layer cutoffs. Locally, the clay smear consists of overlapping patches of sheared clay, separated by sheared sand. More commonly, it is one amalgamated zone of shared sand and clay. Microscopic study of fault zone samples shows that grain-scale mixing can lead to thickening of the low permeability smears, which may lead to resealing of holes.

## 1 Introduction

Clay smears in normal faults are a common feature in layered sediments. Clay smear, (depending on the state of lithification also known as shale smear or –gouge) is entrained along a fault from source beds (Lehner and Pilaar, 1997; Lindsay et al., 1992; van der Zee and Urai, 2005; Vrolijk et al., 2015; Weber et al., 1978; Yielding et al., 1997). This paper deals with unlithified clays and sand and therefore the term clay smear will be used in this work. As clay has a much lower permeability compared to the surrounding sands, clay smears can act as a seal or baffle to fluid flow, especially in a two phase system where a thin clay veneer can hold a large column of hydrocarbons (Smith, 1966; Urai et al., 2008).



The prediction of the sealing potential of a clay smear is mostly based on simple geometrical considerations of the clay content in source beds and the fault throw (e.g. Bouvier et al., 1989; Lindsay et al., 1992; Fulljames et al., 1997; Yielding et al., 1997; Childs et al., 2007). The most common algorithm is the Shale Gouge Ratio (SGR, Yielding et al., 1997), which uses an average of the clay bearing layers thickness and clay content, and the throw. Typical values of SGR at which the clay smear

is thought to be discontinuous are around 0.2 (e.g. Childs et al., 2009; Yielding et al., 1997). A comprehensive summary of clay smear algorithms is given in Freeman et al. (2010). As has been discussed in Vrolijk et al. (2015) these algorithms are based on geometrical considerations, and do not include variations in sedimentary architecture, complex fault structures, or geomechanics. Therefore there is scope for their improvement.

Direct observation of the processes of clay smearing is provided by analogue modeling, numerical modeling and outcrop

studies, all of which are subject of certain limitations. Analogue models (ring-shear, direct-shear, underwater sandbox) provide direct insight into processes and resulting structures and even allow fluid flow measurements on decimeter scales. On the other hand, they are subject to limitations in materials and boundary conditions (Çiftçi et al., 2013; Clausen and Gabrielsen, 2002; Clausen et al., 2003; Cuisiat and Skurtveit, 2010; Giger et al., 2013, 2011; Karakouzian and Hudyma, 2002; Noorsalehi-Garakani et al., 2013; Schmatz et al., 2010b, a; Sperrevik et al., 2000).

Numerical studies using discrete element models (DEM, Egholm et al., 2008) or finite element method (FEM, van der Zee et al., 2003; Gudehus and Karcher, 2007; Nollet et al., 2012; Kleine Vennekate, 2013) are limited to 2D and have limitations due to grain-size and particle numbers (DEM) or the disability to rupture the clay (FEM). However, DEM models show grain-scale mixing and abrasion processes as important factors in clay smear evolution and allow easy parameter studies.

A number of outcrop studies provide information about clay smear in different lithologies and scales. Lindsay et al. (1992)

defined three types of clay smear characterized by the main driving process: (I) shear type, (II) abrasion type, (III) injection type, based on observations in an active quarry (Round O'Quarry, UK).

Weber et al. (1978) and Lehner and Pilaar (1997) presented observations from fresh outcrops in lignite mines in the Lower Rhine Embayment (LRE), Germany. The sediments are not lithified and the clays are soft, as the authors report clay being pressed out of the fresh cuts. Lehner and Pilaar (1997) present a model for the development of injection-type clay smear

consisting of two overlapping fault segments that form a pull-apart structure into which the weak clay can move. This is accompanied by intensive host rock deformation by D-, R- and R'-shears influencing the overall structure of the clay smear. This pull-apart injection model is supported by observations of Clausen et al. (2003) from faults in Bornholm, Denmark, Doughty (2003) from slightly lithified sediments in the Rio Grande Rift, New Mexico, USA, Faerseth (2006) from several large scale faults, van der Zee et al. (2003) and van der Zee and Urai (2005) from outcrops in Miri, Sarawak, Malaysia. Based

on outcrop observations from the Hambach mine, van der Zee et al. (2003) developed the Mechanical Clay Injection Potential algorithm (MCIP) which predicts whether or not a clay injection is possible at a certain setting.

A model for less ductile clays and shales, the shear-type clay smear consists of two vertically overlapping fault segments that successively incorporate and attenuate the clay with ongoing offset (Aydin and Eyal, 2002; Doughty, 2003; Faerseth, 2006; Lindsay et al., 1992).



Burhannudinnur and Morley (1997) noted a distinct mixing of shale and cataclastic fragments in the clay smear at outcrops in northwest Borneo, Malaysia. Grain scale mixing appears to be a common process in faults in poorly lithified sediments (Bense et al., 2003b; Heynekamp et al., 1999; Kristensen et al., 2013; Loveless et al., 2011) and is also noticed in analogue experiments (Clausen and Gabrielsen, 2002; Noorsalehi-Garakani et al., 2013; Schmatz et al., 2010b) and numerical models (TerHeege et al., 2013). Bense et al. (2003b) and Kristensen et al. (2013) additionally noted a reorientation of grains along slip planes. Shale smears in lithified sediments show less or no grain scale mixing, but incorporation of wall rock fragments Aydin and Eyal (2002); Eichhubl et al. (2005); Foxford et al. (1998); van der Zee and Urai (2005). Such fragments are often subject to locally increased shear stresses and thus strongly deformed (van der Zee and Urai, 2005).

Foxford et al. (1998) studied the structure and fault rock content of the Moab Fault, Utah/USA at numerous outcrops. They described a highly variable thickness of shaley gouge (cm to meter scale) and the fault rock in general, concluding that it is impossible to predict content or thickness of the fault zone from observations, although the presence of shaley gouge might be predictable. A critical SGR value of 20 % for the Moab fault is suggested, but the authors note that empirical databases for individual fields are required to implicitly include sub-seismic effects such as throw partitioning Noorsalehi-Garakani et al. (2013); van der Zee et al. (2003).

An excavated fault zone in a sandstone/shale sequence consisting of lenses of clay and sand is described by Childs et al. (1997) from a quarry in Lancashire, UK. Lenses can be clay dominated, sand dominated or mixed and in between the lenses the fault shows the respective wall rock. This study shows the complexity of fault zones with multiple fault strands and the importance of understanding the processes of fault development. Similar shale-rich lens structures of limited extent were described by Vrolijk et al. (2005) within a fault zone rich of relays. These authors discussed the effects of sedimentary architecture such as channels on clay smear evolution (cf. Davatzes et al., 2005).

Doughty (2003) studied clay smears at the Calabacillas fault, Rio Grande Rift, New Mexico, presenting a 3D thickness map of the clay gouge, that was interpolated from numerous measurements. He described several gaps in the gouge that are interpreted to be formed by secondary faults and compromise the fault seal integrity.

The effect of multiple clay layers on clay smear continuity and permeability was studied by Childs et al. (2007) in outcrops of the Taranaki formation, New Zealand. Combined with theoretical considerations it results in the formulation of the Probabilistic Shale Smear Factor (PSSF) that defines the chance of encountering a hole at a certain position along the multilayered clay smear.

Mining in the Lower Rhine Embayment causes large gradients in hydraulic head which are monitored in detail. Bense and Van Balen (2004) used these data along with SGR estimations in numerical groundwater flow models across a relay structure. Results imply that the faults are sealing. Bense et al. (2003a) report on the groundwater flow associated with these faults, suggesting they form baffles for cross fault flow and enhance vertical flow. An outcrop study in the close-by Roer Valley Rift System Bense et al. (2003b) reports that this heterogeneity is a result of disaggregated sand bodies in the damage zone of the fault. Bense et al. (2003b) also propose pebbles in the fault to cause holes in the clay smear and describe how clay smears are enhanced in volume by grain-scale mixing.



In summary, most of these studies are restricted to vertical or horizontal profiles. For faults with displacements of tens of m only relatively small portions of the clay smear can be studied in detail because of outcrop size limitations. It is commonly seen that fault structures and clay smear thickness can vary strongly over short distances, both along strike and dip. Despite the consensus on this complexity the processes and structures associated with variations of clay smear structure and thickness
in 3D are not well understood. Since outcrop studies investigating the entire faulted sequence including clay source beds are scarce we also have little understanding of the transition of source bed to clay smear. One way to improve this understanding is to study clay smear in 3D, integrating data from source beds and the fault zone, finally aiming for a geomechanical model that can explain the observed structure. This is almost non-existent at present, because of the difficulty of finding suitable outcrops where throw values are in the range of feasible excavation depths, SGR values are in the desired range and sediments are soft
enough to allow excavation.

In this paper we present an outcrop study from the Hambach open cast lignite mine, near Cologne, Germany. Here, conditions outlined above are present, providing a world-class site to investigate clay smears. We access faults in sand-clay layers in fresh outcrops which have never been buried deeper than approx. 150 m, and have access to an excavator to prepare 3D outcrops of selected sites. We provide detailed observations of unlithified, faulted deltaic sand-clay sequences in excavated outcrops in 3D,
including thickness distributions, observations of host rock deformation affecting clay smear geometry, the effect of grain-scale mixing and potential clay smear disrupting processes. We then discuss the interplay between structural and mechanical effects and the implications for the evolution of clay smear.

## 2   Geological Framework

The Hambach lignite mine is part of the Lower Rhine Embayment (LRE) in western Germany, (Fig. 1; redrawn from Knufinke
and Kothen, 1997) which formed in the European Cenozoic Rift System (Ziegler, 1992), starting in the late Eocene (35 Ma) till the Pliocene. The thick lignite layers that are mined today formed during the Miocene. Synsedimentary faults that were active from the late Oligocene to Pliocene (Knufinke and Kothen, 1997; Prange, 1958; Quitzow, 1954) play an important role in the LRE, as they control sedimentary architecture and often develop clay smears in the soft sediments with a strong effect on groundwater flow (Bense and Van Balen, 2004; Bense et al., 2003b; Spiller et al., 2004). The LRE is still tectonically
active today (Kübler, 2012; Winandy et al., 2011) and groundwater drainage around the open pit mines causes heterogeneous subsidence and potential movement on faults (Kübler, 2012). However, surface observations of the mining operators indicate no recent activity on the studied "Etzweiler Sprung"-fault.

Faults or fault segments with clay smear in the Hambach mine occur at a wide range of scales. They include pebbles and fragments of brittle lignite that are incorporated into the clay smear and can affect its structure and continuity. Clay layers
range from meter to millimeter in thickness and SGR values can be as low as 0.05. Thick clay source layers are continuous but thinner layers form from rip-up clasts embedded in sand with strong lateral variation. This wide range of structures makes it a perfect place for outcrop studies.



Extensive statistical analysis of 2D outcrops in the Hambach mine were conducted by Navarro (2002). He investigated three fault sections with throws between 4.5 and 40 m and outcrop lengths between 20 and 330 m. Clay smear thickness data show log-normal distributions for all faults. Fault roughness analysis implies that while there is a trend towards a power-law scaling faults do not follow a fractal scaling. Fault sections investigated by Navarro (2002) were located in a thick, brittle lignite layer

and joints in the lignite affected the geometry of the faults distinctly.

Clay smears from larger faults (up to 15-20 m displacement) and thicker source layers in the close-by Inden mine that we excavated were not suited for the main purpose of this study as we could only investigate very small parts of the faults (Fig. 2). However, some important observations came out of these excavations: (1) The clay smear thickness varies distinctly between the investigated sections; (2) the fault zone can be very wide (as expected); (3) clay smears are layered, presumably reflecting

the original stratigraphy, and in some cases a thick zone of sheared sand is found between two clay smears.

## 3  Field Observations

After a preparation phase with observations in selected locations, we focused our study on the fifth floor of the Hambach lignite mine, about 150 m below surface level. The sediments here are at their maximum burial depth without tectonic overprint after the normal faulting. Our aim was to find faults with a low SGR and throw that allows the excavation of both source layers and

the clay smear. This range is predefined to about 1.8 m trench depth by security standards in the mines. If we are looking for faults with SGR < 0.2, source clays have to be < 20 cm thick. The larger faults in the area have offsets of several meters, and in these we thus cannot expose both source layers and the fault zone. Based on existing mine fault maps we selected a relay of the 'Etzweiler Sprung'-fault zone as target of the detailed study as we expect a number of smaller faults in this region.

In this area, we cleared off the top 30 cm of disturbed material with an excavator to expose the main fault with a dip-slip of 8

to 10 m, and located the fault relay in which several synthetic smaller faults in the hanging wall, offsetting thinner clay layers. Outside the fault zones, the study area shows sub horizontal source beds dipping NW with < 5°. An overview of the excavation side is shown in Figure 3. We focused on these smaller faults as they provide entire fault sections including source layers on hanging wall and footwall cutoff. Here, 4 trenches of 1.5 m depth were excavated (map of faults, trenches and surfaces in Fig. 4). The walls of the trenches were cleaned with large, sharp cutting tools to expose the clay gouge and deformation bands

in the sand. The 7 vertical cross-sections were named 1.1 (trench 1 NW section) and 1.2 (trench 1 SE section), 2.1 (trench 2 NW section) and 2.2 (trench 2 SE section) etc. After this, we removed the hanging-wall sand to expose the clay gouge in 3D between section 1.2 and 2.1 (surface 1) and between 3.2 and 4.1 (surface 2). Finally, at surface 2 the outcrop was sliced in 5 cm increments and compiled to a 3D thickness model (see section 3.3).

### 3.1  Excavated Surfaces

Removal of the hanging-wall sand using brushes was possible since the sand is almost cohesionless while the clay is stronger and allows brushing off the sand without being deformed by the bristles. This sharp change in material strength allows a



preparation of clay smear surfaces similar to the experiments of Noorsalehi-Garakani et al. (2013). In this way, we excavated two clay gouge surfaces of 1.8 m$^2$ (surface 1) and 3.8 m$^2$ (surface 2).

At first look, both clay smear surfaces contain many sub-horizontal clay ledges with fine horizontal layering locally visible (ellipses labeled (1) in Fig. 5a and Fig. 7a). Striations on the clay smear surface (dashed lines labeled (2) in Fig. 5a and Fig.

7a) can deviate up to 10° from dip-slip.

Surface 1 (Fig. 5a) is relatively smooth. In profile the clay smear is between 0.5 and 2.8 cm thick, in a wider fault zone of deformed sand with anastomosing deformation bands. The source layer is visible with both hanging wall and footwall cutoff.

At the top center of the excavated surface a truncation of the gouge is visible (transparent red area marked with a star, Fig. 5a). This hole in the smear was created by a mistake during the excavation process. However, we are confident, that the rest of

the excavated surfaces is undisturbed, because the sand separates very easily from the cohesive clay.

The sub-horizontal clay ledges with fine horizontal layering can be locally shown to curve into the clay gouge which is made of multiple sheared clay layers separated by thin sand seams. Figure 6a shows this in detail in the zone marked in Figure 5a illustrating the different clay smear layers, interpreted to have formed from different source beds (cf. conceptual sketch Fig. 6b). The clay smear surface is colored by yellowish iron hydroxides. Black striations (dashed lines labeled (2)) are interpreted

to be the result of the fault moving past lignite fragments. Vertical sections at both sides of the clay smear are shown in Figure 5b & c.

At surface 2 (Fig. 7a) the clay gouge shows a rougher, patchier structure compared to surface 1. Clay patches on the surfaces occur in different shapes and sizes (e.g. transparent green areas). Again the surface shows a reddish-yellow oxidation color and black lignite streaks. The footwall cutoff is located a few decimeters above excavation level and therefore not visible.

Consequently, the displacement is extrapolated from vertical section 4.2 (Fig. 8b), 3 m to the SE, to be 1 – 1.5 m. Vertical sections at both sides of the surface are shown in Figure 7b & c.

### 3.2   Vertical Sections (Profile View)

Faults in the sand start out as deformation-band-type shear bands or disaggregation bands Fossen et al. (2007). Dilation of the sand enhances fluid flow and related bleaching causes the shear bands to appear lighter in color. At offsets above a few cm,

bundles of deformation bands outline the faults. Following the terminology used by Lehner and Pilaar (1997) we observe (1) D-shears, which are following the main fault dip, (2) Riedel- or R-shears, which are oriented in a small angle to the main fault dip, and (3) R'-shears oriented in a high angle to main fault dip. These shears are clearly visible with offsets in the mm to dm range. Where no clay is in the faulted section the entire offset is accommodated by few D-shears and a varying amount of R- and R'-shears. However, once a clay smear is present, shear can be localized entirely within the smear (no deformation bands

in the adjacent sand), cross the smear from hanging- to footwall (bundles of deformation bands in the sand at low angle to the clay smear) or be distributed between the clay smear and parallel D-shears. R- and R'-shears in hanging- and footwall can offset each-other as well as the clay smear as described in the following paragraphs.





Vertical sections show that the anatomy of the studied faults changes over few meters in both stratigraphy and geometry. While section 1.1 (Fig. 8a) consists of up to four small-offset faults (25-50 cm offset) and footwall deformation is minor, at trenches 3 and 4 a larger fault strand with more than one meter offset is dominant (Fig. 7b & c).

Sections 1.2 and 2.1 (Fig. 5b & c respectively) at both sides of surface 1 are fairly similar. In both, source layers in hanging- and footwall are visible, consisting of rip up clasts of different sizes. In both profiles a continuous smear consisting of a mixture of the sand and clay in the source layer developed, with the thinnest part being close to the footwall cutoff. Deformation band density in the footwall is lower than in the hanging wall. SGR is about 0.3. In section 1.2 the thickness of the fault zone is quite variable (boundaries outlined by dashed lines in Fig. 5b & c) and is thinnest in the upper part and thickest in the middle where coincidentally the clay smear is thickest, while in section 2.1 shear appears to localize at the boundary of clay and sand and bedding is horizontal up to the clay smear. The clay smear in section 2.1 is overall thicker and shows no irregularities.

Sections 3.2 and 4.1 at both sides of surface 2 (Fig. 7b & c respectively) differ distinctly from each other and from the sections in trenches 1 and 2. Section 3.2 consists of 2 clay smears composed of deformed and coalesced rip-up clasts with a layer of sand sheared in between. The total offset cannot be determined exactly, but the visible source layer thicknesses and offsets for both individual clay smears suggest SGR values higher than 0.2. While the footwall does not show strong signs of deformation, the sand in between the source layers is strongly deformed, with a high density of R-shears that terminate at stair-stepping structures in the upper clay smear.

Section 4.1 on the SE side of surface 2 consists of one visible clay smear (Fig. 7c). The hanging wall cutoff shows one source layer with an estimated SGR of 0.05. The footwall is divided into two zones by different deformation mechanisms. The upper part is dominated by R- and R'-shears and one D-shear on the footwall side of the clay smear. R- and R'-shears terminate in stair stepping structures at the footwall side of the clay smear. The hanging wall side of the clay smear is comparably smooth. In the lower part of the section R- and R'-shears are absent. Instead a wider shear zone indicated by sheared lignite seams developed. The hanging wall side of the clay smear is now rougher and we observe a higher sand content in the smear, indicating a stronger grain-scale mixing. Over the entire section the clay smear thickness varies, however, we do not observe disruptions of the smear. A detailed interpretation of this section is shown in Figure 13 and discussed in section 5.1.

Two additional cross-sections are shown from different field campaigns in the same mine and at the same fault and level. Cross-section 5 in Figure 8c (see Fig. 15 for interpretation) shows stair-stepping structures at the footwall side of the clay smear and numerous R-, R'- and D-shears forming two deformed clay smears. The cross-section shown and interpreted in Figure 22a & b, respectively shows a thicker clay smear with brittle fragments of lignite in it.

### 3.3 Clay Smear Thickness Distribution

The cleaning procedure has shown that over the two cleaned surfaces the clay smear is continuous (a gap where the clay is absent would have shown up clearly by the brushing procedure.

At the north-western part of surface 2 we incrementally cut 13 sections with a distance of 5 cm and took high resolution photos of each (examples shown in Fig. 9a-c). These photos are loaded into a GIS software and scaled to fit a common reference system. Then the footwall and hanging-wall contact between clay smear and host sand is interpreted with sampling distances



of approximately 0.5 cm. A clear boundary exists between pure sand, sand-clay mix or pure clay in the fault, so that there is not much interpretation required for the digitizing. To extract the thickness of the clay smear, XY-data of both digitized traces are transferred to MATLAB (2015), where we rotated the traces to a horizontal orientation, interpolated the digitized sampling points so that both data points in both traces have the same X-values, and then calculated the clay smear thickness (clay and mixed sand-clay). Data from all 13 sections are finally interpolated to a thickness map using the integrated interpolation algorithm ('interp2') of MATLAB (2015). A graphical representation of the workflow is shown in Appendix A.

The thickness map (Fig. 9e) shows that the clay smear is patchy, with a gradual change between profiles. A general trend is towards thinner clay in lower left, with but horizontally elongated thicker patches are distributed over the entire area. Thick clay smear is located in the lower central part as well as the upper 50 cm of the smear. However, even close to the footwall cutoff of the source clay, thin clay smear (less than 1 cm) occurs in sections 3, 9 and 10. The excavated clay smear surface (Fig. 7a) as well as the sections show that the hanging-wall side of the smears is relatively flat, while the footwall side is defined by numerous stair-step asperities. A large sample of the clay smear (location shown in Fig. 7a) shows the 3D geometry of these stair-steps in detail (Fig. 10a). A photogrammetric 3D model created using Autodesk® 123D® Catch software is provided as online supplement (Fig. 10b, image created using Meshlab (Cignoni et al., 2008). These stair-steps create sudden changes in clay smear thickness and determine part of the thickness distribution. The measured thickness data show a log-normal distribution (Fig. 11) similar to those shown by Navarro (2002) from 2D profiles.

## 4   Samples

Samples were taken as blocks from around the footwall cutoff of surface 1. Sample 1.2 consequently is a block taken from the left side of Surface 1 and Sample 2.1 from the right side. Both samples were slowly dried and casted in resin to allow a sawing of the samples without damaging the clay to provide surfaces that clearly show sand grains embedded in the clay smear. Very high resolution photography then allows studying the samples on sand grain size scale.

In Sample 1.2 (Fig. 12a & b) the source layer consists of rip-up clasts in mostly sub-cm scale with distinct amount of sand in between. This sand is sheared and dragged into the shear zone along with the clay, where we observe a composite clay smear consisting of sand and clay. Several thin slip planes with minor displacement extend through the footwall side of the source layer with some distance to the main shear zone (dashed lines in Fig. 12a). At the top of the sample, i.e. the very footwall cutoff on the hanging-wall side we note the highest sand content that decreases further towards the footwall.

The source layer in Sample 2.1 (Fig. 12c & d) shows one thick clay clast of 5-8 cm thickness with a 1.5 cm thin layer of small rip-up clasts on top. This layer of rip-up clasts is entrained into the shear zone and progressively mixed, resulting in a gradient of sand content, increasing towards the hanging wall.





## 5  Discussion

### 5.1  Origin of stair-stepping geometries in clay smear

The studied faults are part of a larger fault system and are interpreted as synthetic faults in a relay of a larger fault. The position in the relay is interpreted to locally enhance footwall and hanging-wall deformation by R-, R'- and D-shears at some but not
all faults. Where wall-rock deformation occurs, it strongly affects the shape and continuity of the developing clay smears.

The interpretation that R- and R'-shears are an essential part of fault zones with clay smear has been proposed by Weber et al. (1978) and Lehner and Pilaar (1997) in similar outcrops of the LRE. They observed R-shears forming horse structures at the hanging wall side of the clay smear. These steps in the lower part of the clay smear are the result of a process that van der Zee and Urai (2005) called telescoping, a stepwise migration and shallowing of fault segments towards the hanging wall. However,
effects of R'-shears on clay smear are not visible in the outcrops of Lehner and Pilaar (1997). In contrast to their observations, our outcrops show rarely any clay smear deformation on the hanging-wall side but numerous stair-steps, always at the footwall side of the clay smears. Eichhubl et al. (2005) interpreted these structures as result of R-shears in an intermediate stage of clay smear evolution after which slip localizes sharply following the general fault dip. Similar structures were described by Çiftçi et al. (2013) and Giger et al. (2013) in crosscuts of analogue clay smear models. In the following we present three models
which can form stair-stepping structures in the clay smear and are thus responsible for the thickness distribution shown in this article's section 3.3.

(A) In section 4.1 (see Fig. 13b for interpretation) we observe late R-shears offsetting older D- and R'-shears on the footwall side, terminating in the clay, and closely associated with the characteristic stair-steps (Fig. 14a). Therefore our first model involves a highly strained clay smear, with the R shears nucleating on the footwall side. This is combined with a redistribution
of the shearing clay to maintain continuity. Where R-shears truncate the clay smear it locally becomes very thin. This process may be able to form holes in a clay smear with enough offset on the R-shears (cf. paragraph 5.4).

(B) We observe R'-shears causing stair-steps in clay smear as well (section 5, Fig. 15). Here, the process is analogous to that described in (A): in a first step faults with small displacement develop clay smears and cause the formation of D-shears around the smears (Fig. 15a-d). In a next stage R'-shears develop in the footwall in low angle dips (Fig.15e). These
25 R'-shears with 1-2 cm displacement offset the earlier formed D-shears. While growing through the footwall they cut through the undeformed source clay beds which provide weak slip zones and allows the R'-shears to be almost horizontal. Finally, the R'-shears protrude towards the main clay smear offsetting the clay-smear/sand contact in the footwall and hence forming triangular stair-steps (Fig. 14b). As in model (A) the clay in our outcrop has to be plastically redistributed within the smear to maintain continuity. Kristensen et al. (2013) reported a clay smear that was disrupted by R'-shears with offsets larger than the
30 clay smear thickness.

(3) Clay lenses incorporated in the fault by fault segmentation can get eroded on the hanging-wall side with continuing shear (Fig. 14c). This results in a straight surface on the hanging-wall side, while on the footwall side a step remains (cf. analogue models of  Noorsalehi-Garakani et al., 2013; Schmatz et al., 2010b, a). This process forms stair-stepping geometries without R- or R'-shears.



A simple criterion to test if a clay is able to be injected is proposed by van der Zee et al. (2003), the mechanical clay injection potential (MCIP) defined as MCIP=$\frac{\sigma'_1(1-sin\phi)}{(2cos\phi)C}$, where $\sigma'_1$ is the effective maximum principle stress, $\phi$ is the friction angle and C is the cohesion of the clay. A MCIP > 1 indicates a possible clay injection. Clays in the 5th floor of the Hambach mine have cohesions of around $C_{min}$ = 30 kPa to $C_{max}$ = 90 kPa and friction angles of $\phi_{min}$ = 9° to $\phi_{max}$ = 14° (pers. comm. Prof.

Dr. D. Dahmen, RWE Power AG). The 150 m overburden has an estimated average density of 1937 kg/m$^3$ resulting in $\sigma'_1$ = 2.8 MPa. The MCIP then calculates to MCIP$_{min}$ = 5.9 and MCIP$_{max}$ = 19.6. This means that the stress-strength relation can allow injection and it is likely that within the faults ductile redistribution of the clay can occur.

As stated before there are several empirical methods to predict the sealing potential of a clay smear, which are based on their actual deformed configuration. These methods are often criticized because they overlook the mechanical and hydraulic

behavior of the sealing material. Based on numerical simulations Kleine Vennekate (2013) proposed a new methodology to evaluate the continuity of the clay smear in a normal fault. This methodology takes into account not only the source clay thickness and throw but also considers the stress state and the shear strength of the low permeable layer.

The evaluation of the stress state and shear strength follows the idea of the MCIP proposed by van der Zee et al. (2003), which infers if the clay deformation occurred in a tension or compression regime for the lowest principal stress ($\sigma_3$). This is

15 achieved by assessing two angles $\beta$ and $\alpha$ in a principal stress $\sigma_1$ and $\sigma_3$ diagram, which relate the assumed stress path during the deformation and the shear strength to the stress state of the clay layer before the deformation. A ratio $\frac{\beta}{\alpha} < 1$ implies that $\sigma_3$ will be negative during the deformation, otherwise $\sigma_3$ will be positive. The deformed configuration is considered by using the shale gouge ratio. Both criteria, $\frac{\beta}{\alpha}$ and SGR, are then plotted together with a curve that marks the limit between a continuous and discontinuous clay gouge.

This methodology was used to assess the continuity of the clay gouge in the excavated fault. Figures 16 (a) and (b) show in a principal stress diagram ($\sigma_1$, $\sigma_3$) the Mohr-Coulomb limit state line and the assumed stress path together with the angles $\beta$ and $\alpha$ for c = 30 kN/m$^2$, $\phi$ = 9° and c = 90 kN/m$^2$, $\phi$ = 16°, respectively. The calculated ratio $\frac{\beta}{\alpha}$ can vary approximately from a value of 8 up to a value of 28, implying that the value of $\sigma_3$ was positive during the deformation.

The continuity of the clay smear is then evaluated in Figure 17. Here the limit between continuous and discontinuous clay

smear is presented with a continuous line and the calculated upper and lower limit of both SGR and $\frac{\beta}{\alpha}$ are plotted with dashed lines. The shaded area represents all possible combinations of SGR and $\frac{\beta}{\alpha}$. According to Kleine Vennekate (2013) it can be expected that the clay smear would be continuous since this area is above the limit curve, the continuous clay smear zone, which seems to agree with the field observations.

## 5.2 Evolution of layered clay smear

In places where the distance between two clay beds is large enough and the sand fails in shear (e.g section 3.2, Fig. 7b), the interbedded sand layers are sheared along with the clay smears and finally amalgamate to form a continuous 'sand smear'. Based on simple mixture theory arguments, the optimum for this is around 35 % clay in the sheared section (Crawford et al., 2002), where the porosity is the minimum with many thin clay layers more prone to amalgamation than a few thick ones. An





important observation was that some of the clay smears which we were able to mechanically dissect into individual clay bands separated by sheared sand (e.g. surface 2, cf. Fig. 6) did in fact look rather continuous in profile. We distinguish two cases:

1. Amalgamated clay smears may have patches consisting of 'masonry-like' stacks of clay in sand - with a tortuous sand path across the clay smear. Across- and along-fault flow is possible, but baffled due to the high tortuosity.

2. Individual layers of amalgamated clay smears may be continuous but maintain a thin sand veneer in between. An effective conduit for along-fault fluid transport can be maintained, while across-fault flow is hindered.

Aydin and Eyal (2002) suggested that clay layers merge when the throw is larger than the distance between clay layers. In their case however, 'brittle' sands between softer clays do not form a 'sand smear' but are rather boudinaged sand fragments embedded in a composite clay smear. From this distinctly different behavior of faults it becomes clear that mechanical properties of sand and clay are of great importance for the evolution of faults with clay smears. Vrolijk et al. (2015) discussed the characteristic features and processes to be expected in different sand-clay strength ratios (cf. strength matrix, their Fig. 27). Following this classification the outcrops described by Aydin and Eyal (2002) plot in the lower left quadrant of Vrolijk et al.'s (2015) strength matrix, where sands fail in extension and clays are weaker than the sands.

However, section 3.2 (Fig. 7b) of this study clearly shows that clay smears do not merge with offsets exceeding the distance between source layers by far. This was also observed in cm-scale outcrops by Kristensen et al. (2013) and analogue sandbox experiments of Schmatz et al. (2010a). Both, our outcrops as well as the analogue models, are defined by sands that fail in shear and clays that have a comparable strength, thus plotting in the upper left quadrant of Vrolijk et al.'s (2015) strength matrix.

Aside from mechanical properties of sand and clay, geometries are of importance as well. Whether or not clay smears merge depends on the distance between source layers, fault dip and shear zone width.

Childs et al. (2007) developed a statistical approach to estimate the probability of breaching a multilayer clay smear based on outcrop observations, the Probabilistic Shale Smear Factor (PSSF). This approach calculates the probability to encounter a hole in the clay smear at a specific location by investigating the same probability for each individual smear. Our observations however, imply that in case of multilayered clay smears with comparable strengths of sands and clays where both fail in shear it is not required to have a disruption of all individual smears at the same position. Continuous sand smears can connect holes in different locations of the smears even at large displacements allowing across-fault flow. Therefore, PSSF requires discontinuous sand layers between the clay smears. While for the given material strengths a very high offset can lead to a complete mixing of the sand into the clay smear, this is best achieved by strength-ratios plotting in the lower left quadrant of Vrolijk et al.'s (2015) strength matrix.

## 5.3 Continuous clay smear from discontinuous source layers

In the Hambach outcrops which are mostly composed of fluvio-deltaic sediments the thinner clay source beds consist of clay rip-up clasts (cm to dm scale), possibly formed in tidal channels, embedded in a sand matrix. These layers can thus be quite permeable in undeformed state. Shear deformation during faulting strongly elongates the clasts and grain-scale mixing increases the fraction of sand-clay smear, as shown in sample 2.1 (Fig. 18a). The high content of sand in the smear in turn





changes the strain partitioning as a higher sand fraction increases the residual friction angle Lupini et al. (1981) and shear strength Vallejo and Mawby (2000). Thus, smears from individual clasts do not survive but are sheared, mixed and intermingled, and with increasing displacement the clay smear continuity increases, with occasional larger sand lenses enclosed in the clay smear (Fig. 18b). A similar process of mixing and amalgamation of clay smears is inferred to cause a resealing of holes in

individual smears (cf. mixing simulation, section 5.4). Assuming two or more clay smears, each of which has a hole at different locations getting sheared and mixed strongly, holes in one layer can ultimately be resealed by clay from another layer that is continuous at the specific location, when the increasingly tortuous sand becomes discontinuous. It is interesting to note that the rate of shearing, besides elongating clay fragments, may well contribute to the rate of mixing, leading to interesting and yet unexplored couplings.

This observation fits well with the mechanical classification of Vrolijk et al.'s (2015) strength matrix). On the other hand, clay that is stronger than the surrounding sand and fails in extension produces clay fragments which are entrained in the shear zone. Then they can be abraded, so that continuous mixing with host sand and amalgamation of sheared clay fragments ultimately form a continuous clay smear. In terms of fault seal development this implies that faults penetrating through clay beds that are stronger than the surrounding sand can be full of holes at small displacements but reseal with further offset. This process was

observed by Schmatz et al. (2010a) in analogue experiments with cemented source clay bed (cf. their Fig. 10 and Holland et al., 2006).

## 5.4   Grain-scale mixing

Grain-scale mixing as an important process in clay smear development is mainly observed in small scale faults and experiments. In the presented outcrops we observe intense mixing predominantly at two structural elements of the faults: (1) at the hanging

wall side of the footwall cutoff and (2) at sand lenses that are either entrained into or deformed within the smear. To induce mixing between clay and sand grains a shearing at the interface between both is necessary. This is always the case at the footwall cutoff, where we observe an increase in sand content towards the outer part of the clay smear (e.g. Fig. 12b). This process of clay abrasion and mixing was also observed and described by dynamic observations in sandbox models by Noorsalehi-Garakani et al. (2013). They additionally observed that mixing mostly occurred at the hanging wall side of the clay smear, which is in

accordance with our observations.

Mixing causes an increase of the total clay rich volume and decreasing permeability as the clay fills space between sand grains (Bense et al., 2003b; Crawford et al., 2002; van der Zee, 2002). A thickening of clay smears by mixing is also described by Schmatz et al. (2010b), Noorsalehi-Garakani et al. (2013) and Clausen and Gabrielsen (2002) from analogue models as well as in discrete element models by TerHeege et al. (2013). In terms of permeability, a mixed clay smear can be considered a better

seal as long as the clay volume exceeds the pore volume of the sand grains. With even lower clay content the permeability of the mixture increases (Daigle and Screaton, 2015). An interesting observation is that the total volume of the sheared material decreases (clay goes into the pores of the sand) which may have important and yet unexplored consequences for the mechanics of the system.



Careful comparison with the analogue and numerical models discussed above is consistent with our field observation that the main source of sand mixed into the clay smear is located at the footwall cutoff, where the source clay is abraded by the sand. We found no evidence that the amount of sand in the smear increases with further displacement. This localized process of sand incorporation into clay smear that the rate of mixing scales with the amount of sand-clay layer interface; for the same amount strain and of clay in the faulted section, many thin source beds will mix faster than a few thick layers.

To explore the effect of clay fragment size and rate of mixing on the evolution of sand-clay gouge, we designed a simple simulation (Matlab, 2015) code online supplement) where circular clay fragments in a sand matrix are subject to homogeneous simple shear (Fig. 19). With increasing shear strain $\gamma$ a sand-clay mixed seam around the fragments develops and increases in thickness. We ran five series of simulations with initially circular objects representing clay fragments. The rate of mixing is defined as $m = \frac{\Delta T}{\Delta \gamma}$, where $\Delta T$ is the change in thickness of the mixed seam per unit shear strain and $\Delta \gamma$ is the change in shear strain. The thickness of the mixing seam at a given shear strain is then $T = \gamma \cdot m$. Simple shear is then applied to the model and shear strain is increased in steps of 0.05. This was done for five distances between clay fragments (0.1, 1, 2, 5 and 10 cm radius) and four rates of mixing (0.001, 0.01, 0.1 and 0.5). Using the 'intersections' algorithm (Schwarz, 2010) the code finds the strain at which the ellipses intersect (i.e. clay fragments touch) While a mixing rate of 0.5 is certainly unrealistically high, it serves well for illustrating the procedure (Fig. 20). The results show a logarithmic relation between rate of mixing, distance between particles and the strain required to produce an effective seal by mixing (Fig. 21). The initial packing will have an influence on the required strain as well as the distribution of mixing (e.g. stronger mixing at the top of clay fragments), however this will be subject of further research. When the rate of mixing is 0, initially separate clay fragments will never touch.

Other authors report that mixing has only a minor effects on clay smear development: in the small faults described by Kristensen et al. (2013) mixing is minor, while Giger et al. (2013) report that no mixing occurs in their direct shear experiments. However, in our samples (Fig. 12) as well as in analogue models (Schmatz et al., 2010b) we clearly observe grain-scale mixing as important process for clay smear evolution. Quantifying the rate of mixing requires further study and micro-scale analyses, as it will clearly depend on the clay brittleness (Ingram and Urai, 1999; Vrolijk et al., 2015), and microscale deformation mechanisms.

Summarizing these observations, we propose that grain-scale mixing is a process that has its importance in small scale faults with low SGR values and for weak sand and clays. For sands that are stronger than the clay and for larger scale faults Noorsalehi-Garakani et al. (2013) propose an incorporation and subsequent shearing of brittle fragments (cf. also van der Zee and Urai, 2005) to be an equivalent to grain-scale mixing. Some of our observations support this hypothesis as cross-sections including brittle lignite fragments clearly show a transport of the fragments into the clay smear (e.g. Fig. 22), similar to pebbles entrained into clay smear reported by (Bense et al., 2003b). However, the importance of grain-scale mixing on large faults with thick clay smears is largely unexplored and bears potential for future research.

## 5.5 Clay smear termination

SGR values of the investigated faults are often below 0.2 and could therefore be discontinuous according to literature (e.g. Yielding, 2002). In the profiles and excavated fault surfaces we found small holes in clay smears twice, in a vertical (Fig. 23)





and a horizontal profile (Fig. 24). In both cases we interpret secondary faults offsetting the clay smears to be responsible for the holes. This observation is in agreement with the strong effect of R- and R' shears on the clay smear shown in section 5.1.

Attenuation and tapering of the clay smear has been proposed to cause holes in the clay smear in the CSP model. This structure was clearly absent in our observations. The thickness map of surface 2 shows the thinnest parts at the hanging wall

cutoff and thickest parts closer to the footwall cutoff (Fig. 9). Many other profiles (e.g. 1.2, 2.1 or 4.1) show an opposite thickness distribution with thinnest clay smears at the footwall cutoff. Thicker clay patches related to footwall deformation are distributed over the entire clay smear volume (Fig. 9). Thickness maps Çiftçi et al. (2013) produced from direct shear analogue models using CT-scans are in agreement with this observation.

In summary, our observations show that for weak sands and clays thickening of clay smears due to grain-scale mixing is

10 more important than a strain related attenuation and termination. In evolved clay smears, holes in the clay smear are rather the result of secondary faults offsetting the clay smear or initial brittle failure of the source clays (and hence not producing a clay smear in the first place), rather than disruption of the clay smear due to shearing.

## 5.6 Upscaling to larger faults

Upscaling of observations to towards larger faults is based on the idea that faults show a self-similar geometry. Navarro (2002)

studied lateral thickness distribution and geometries of clay smears in lignite in the Hambach mine on faults between 4.5 and 40 m throw. He reported a tendency towards power-law scaling of the faults roughness, but also noticed a deviation from fractal scaling. Torabi and Berg (2011) came to the same conclusion compiling data from numerous fault studies on different scales. They propose the existence of critical displacements which define boundaries of a hierarchal distribution of faults. One reason for this behavior is found in fault interaction and linking. While this variation on different scales can have an effect on some

attributes discussed in this paper (e.g. fault core thickness, see later in this section), other attributes such as the influence of host rock deformation (i.e. R- and R'-shears) are not well studied on different scales but may show a power-law scaling. Further, the fault attributes discussed in this paper are strongly controlled by the mechanical properties of sand and clay and the contrast between both (Vrolijk et al., 2015). Interaction between both lithologies is similar for small and large faults given that the size of the studied system is sufficiently larger than the grainsize.

Therefore, some of the observed clay smear processes can be upscaled to larger systems directly. We propose that influence of R- and R'-shears on the clay smear structure and potential to form holes is similar in large faults when the relation between SGR and shear-zone width is in the same order or may be even more important as mechanical stratigraphy causes a more complex fault zone. Eichhubl et al. (2005) for instance reported the same stair-stepping structures at the footwall side of a clay smear as shown in this study at a fault one order of magnitude larger. Sufficiently detailed outcrop data of clay smears in

seismic scale faults, however, are lacking and upscaling towards these is tentative at best. Outcrop studies on large faults as provided by Faerseth (2006) or Aydin and Eyal (2002) show similar structures and processes as smaller faults and therefore provide a basis to transfer observations.

Grain-scale mixing cannot be upscaled directly, as grain sizes are the same for small and large faults. Since grain-scale mixing requires shearing at the sand-clay interface a tens of cm thick clay smear may be much less affected by mixing than



mm to cm thick one. An important factor here is the initial rapid increase of fault zone thickness with offset (Torabi and Berg, 2011). Numerous thin source clays contributing to a clay smear are expected to be more prone to grain-scale mixing as one thick clay layer, enhanced by an increased shear zone width resulting from mechanical stratigraphy (Schöpfer et al., 2006; van Gent et al., 2010). In addition, as proposed in the previous section, mixing in larger faults may occur in stages, first by mixing

of sand and clay rock fragments followed by grain-scale mixing by further shear. In summary, mixing is an important process in the evolution of clay smears, but it needs much further study.

We report observations for faults in this study that are one order of magnitude smaller, thus we hypothesize that detailed observations on small scale faults can be transferred to faults at least one order of magnitude larger.

## 6 Conclusions

We present a 3D study of an excavated normal fault with clay smear, together with both source layers, in unlithified sand and clay of the Hambach open cast lignite mine in Germany. The faults formed at a depth of 150 m, and have Shale Gouge Ratios between 0.1 and 0.3. The fault zones are layered, with sheared sand, sheared clay and tectonically mixed sand-clay gouge. There are a few small holes in the clay smear.

The thickness of clay smears is strongly controlled by deformation bands in the footwall. Where deformation bands cross the

15 clay smear they can create holes. Thickness of clays smear in two excavated fault zones of 1.8 m$^2$ and 3.8 m$^2$ is approximately log-normal, with values between 5 mm and 5 cm. The 3D thickness distribution is heterogeneous.

We show that clay smears are strongly affected by R- and R'-shears, mostly at the footwall side. These shears can locally cross and offset clay smears, forming holes in the clay smear, while thinning of the clay smear by shearing in the fault core is less important. Thinnest parts of the clay smears are often located close to source layer cutoffs.

Models of tapering of the clay smear with increasing distance from the source layers are not supported by our observations.

Commonly clay smear is one amalgamated zone of shared sand and clay. Layered clay smears come in two types, one with continuous sheared sand between two clay smears providing vertical pathways for fluid flow, and one which consists of overlapping clay patches separated by sheared sand that provide a tortuous pathway across the clay smear.

Grain-scale mixing is an important process for the formation of continuous clay smears from clay fragments embedded in

sand. This causes clay smears to thicken and reduces permeability. First results from a simplified model suggest that the shear strain required for two clay fragments to connect via shear and grain scale mixing is a logarithmic function of the distance between clay fragments and rate of mixing.

Our results, in agreement with some earlier studies show that fault geometry, layer architecture and mechanical properties all play an important role in the evolution of clay smear.




## Appendix A:  Workflow to determine clay smear thickness from cross-sections

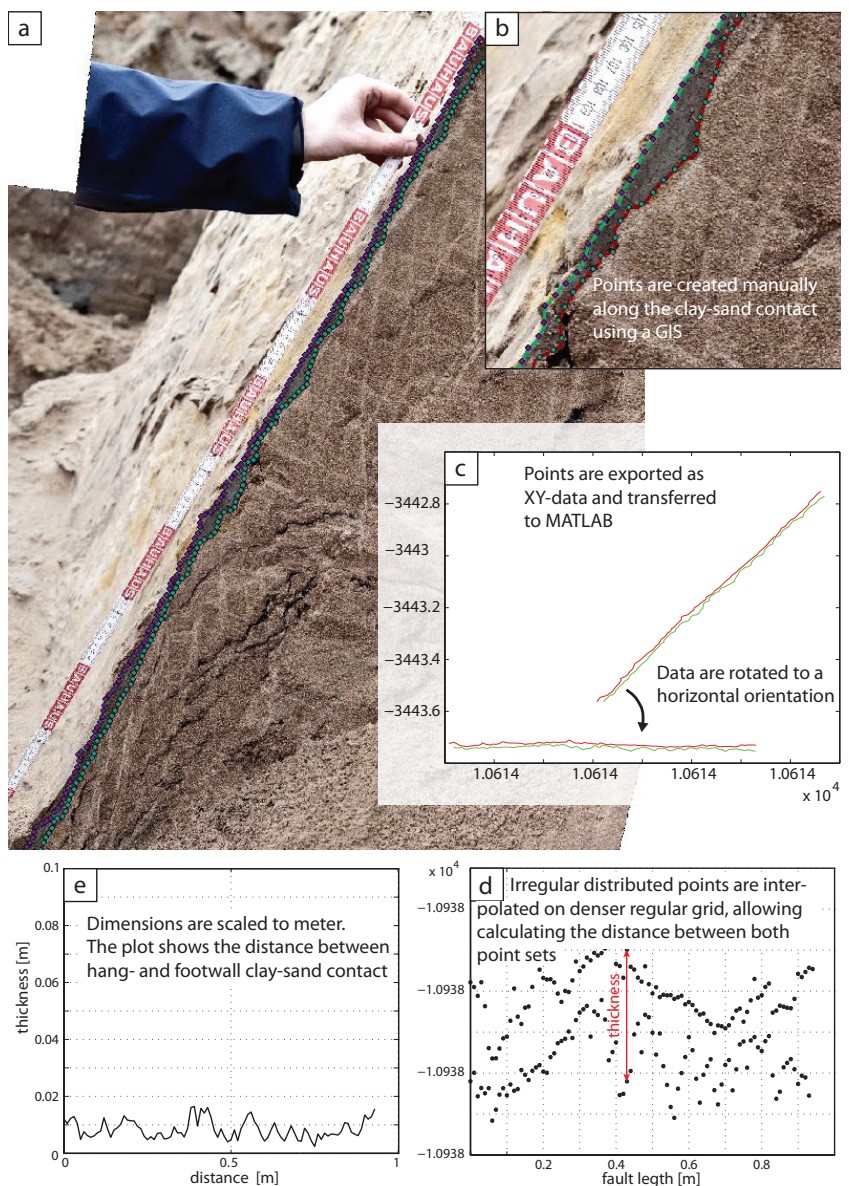

**Figure A1.** Clay smear thickness is determined from orthogonal field photographs (a) of cleaned cross-sections. (b) Clay-sand contacts are manually digitized using a GIS software. (c) Data are then transferred to MATLAB and rotated to a horizontal orientation. (d) Scattered data points are interpolated onto a denser grid that allows calculation of the distance between hanging-wall and footwall data points. (e) Calculated distances resemble clay smear thickness after a final scaling. The plot shows clay smear thickness along the fault.



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

*Acknowledgements.* We thank Peter Lokay, Christian Herberth and their colleagues from RWE Power AG for their kind support during the preparation and realization of the field work and Prof. Dr. D. Dahmen for providing the strength data of the clay. We also thank Sohrab Noorsalehi-Garakani and all students who helped in the field.



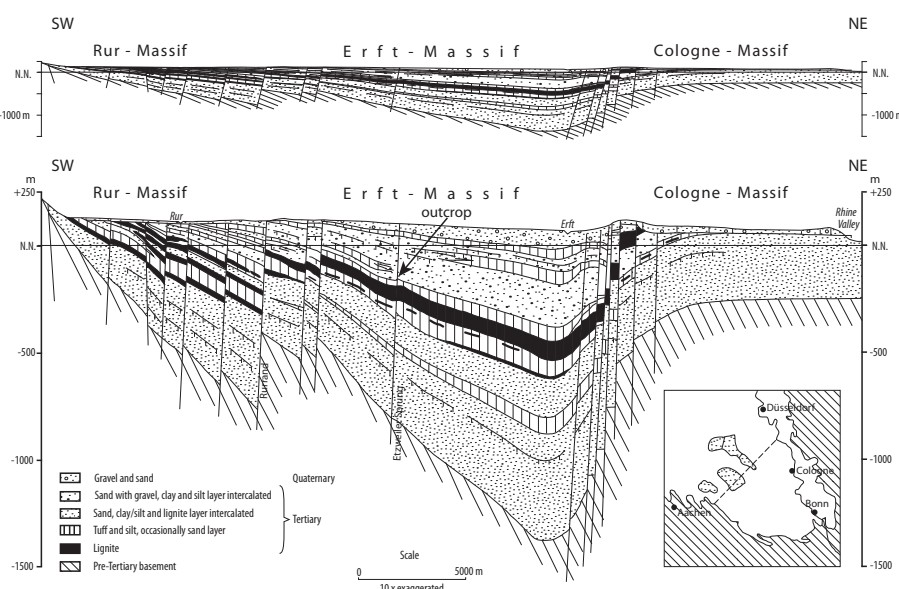

**Figure 1.** SW-NE profile of the Lower Rhine Embayment including the Etzweiler Sprung. An arrow points towards the location and depth of the outcrop. Modified after Knufinke and Kothen (1997).



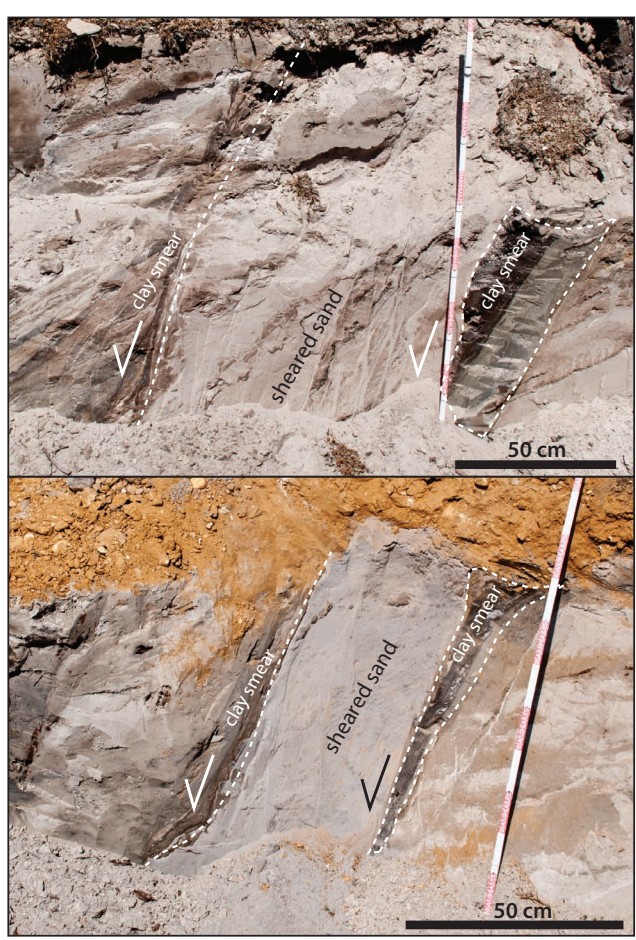

**Figure 2.** Two vertical sections of the Altdorfer Sprung fault with clay smear in the Inden lignite mine. Displacement is about 15 m, the source clays cannot be specified. Distance between both sections is only few m. Note the wide shear zone with thick entrained sand layer as well as the variation of clay smear thickness.




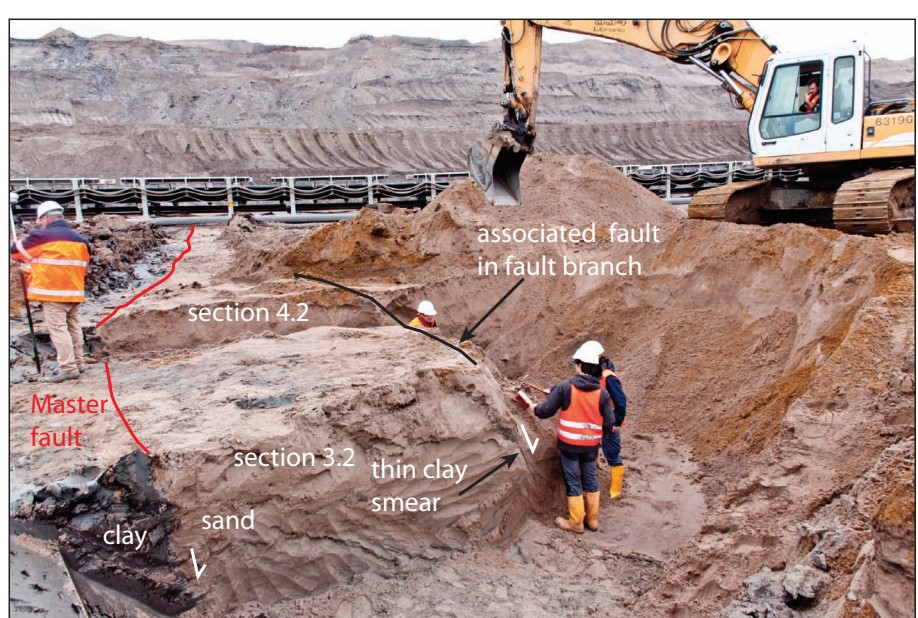

**Figure 3.** Overview showing the excavation side. The master fault is depicted in red, while a black line shows the associated synthetic fault at which the clay smears were excavated. Trenches 3 and 4 are visible.





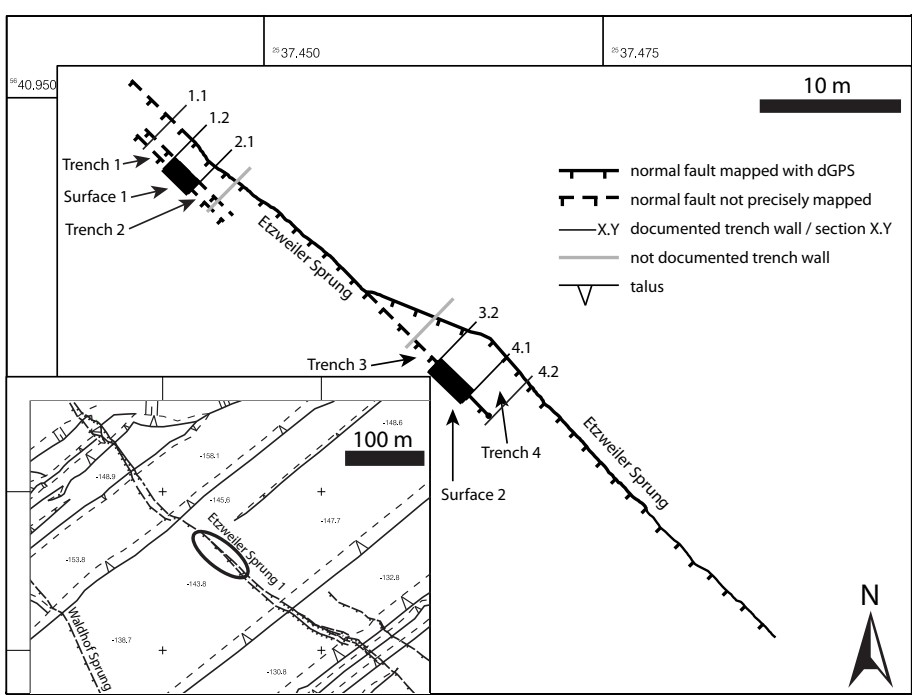

**Figure 4.** Fault map of the outcrop different scales. Trenches, vertical profiles and excavated surfaces are indicated and numbered. Maps provided by RWE Power AG. Continuous fault-line in large map derived from differential GPS measurements. Dashed lines show location of not precisely defined fault traces.



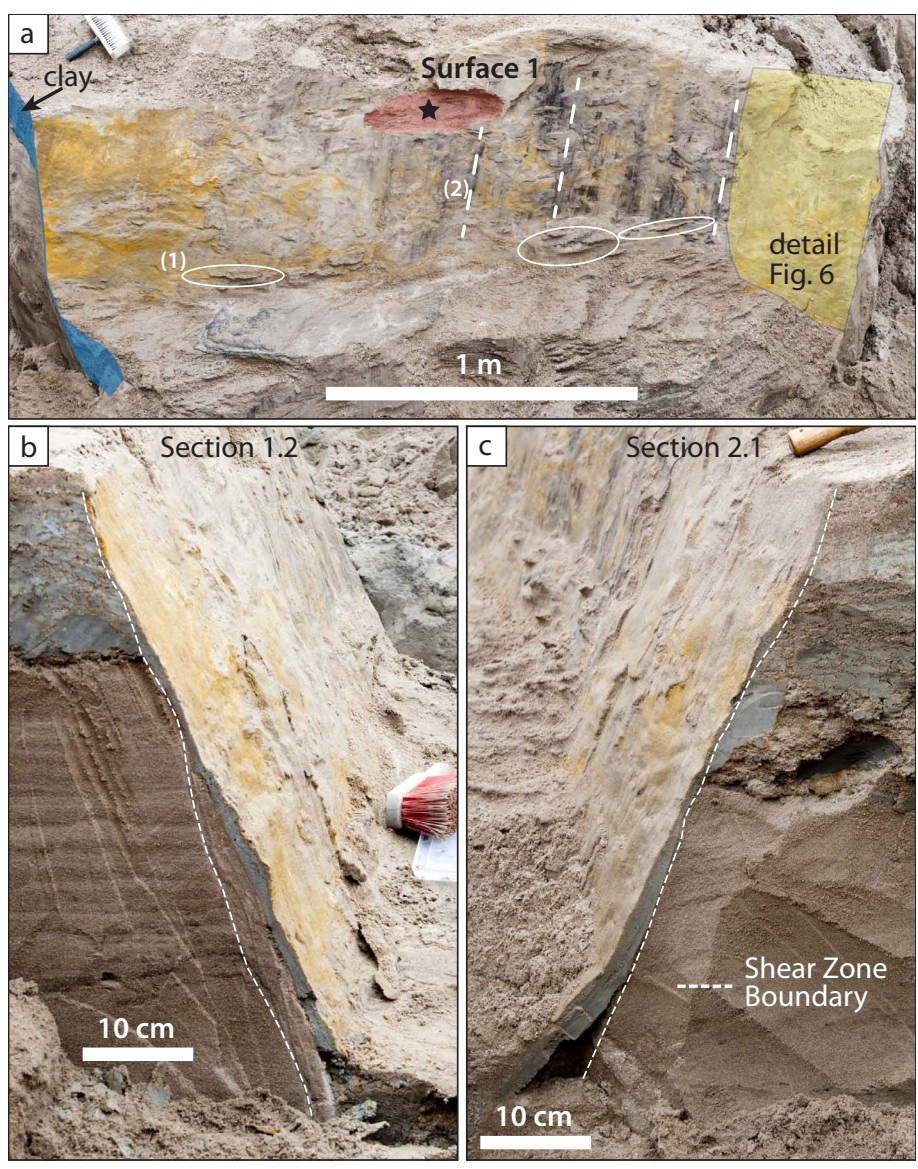

**Figure 5.** (a) Excavated surface 1 and corresponding vertical profiles 1.2 at the NW side (b) and 2.1 at the SE side (c). Shear zone boundaries are indicated by dashed lines in the profiles. Note the thickness variation of the clay smear, with the thickest part in the middle of profile 1.2.





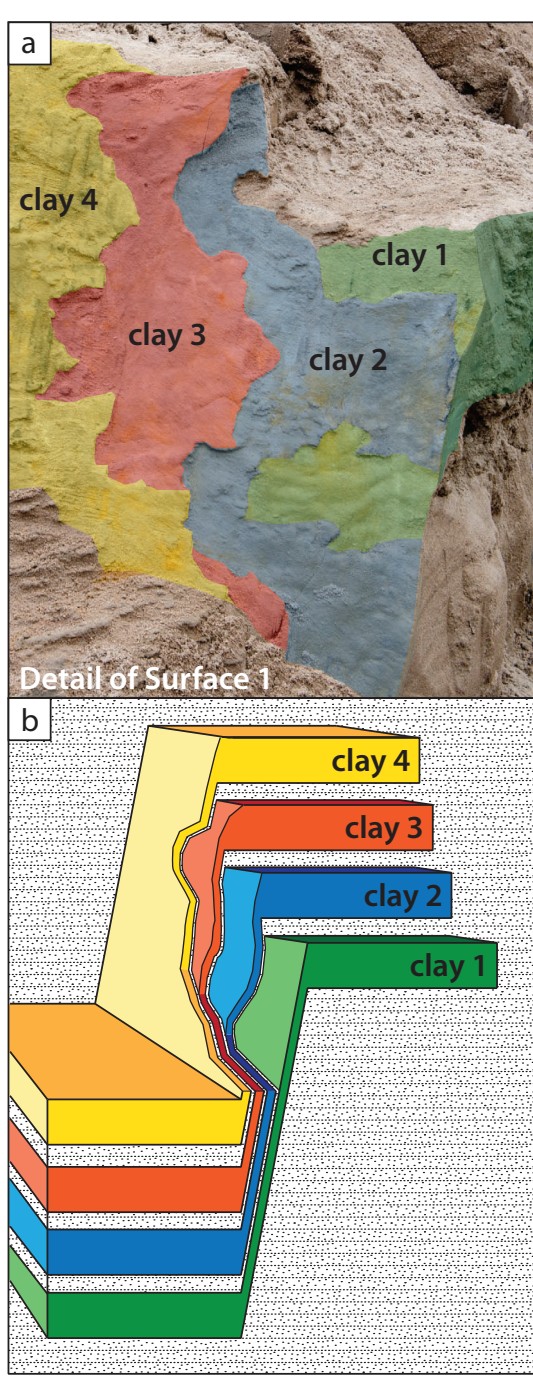

**Figure 6.** (a) Detail of the SE side of surface 1 showing multiple thin clay veneers composing the bulk clay smear. (b) Sketch illustrating the layering of clay smear with interbedded sand layers.





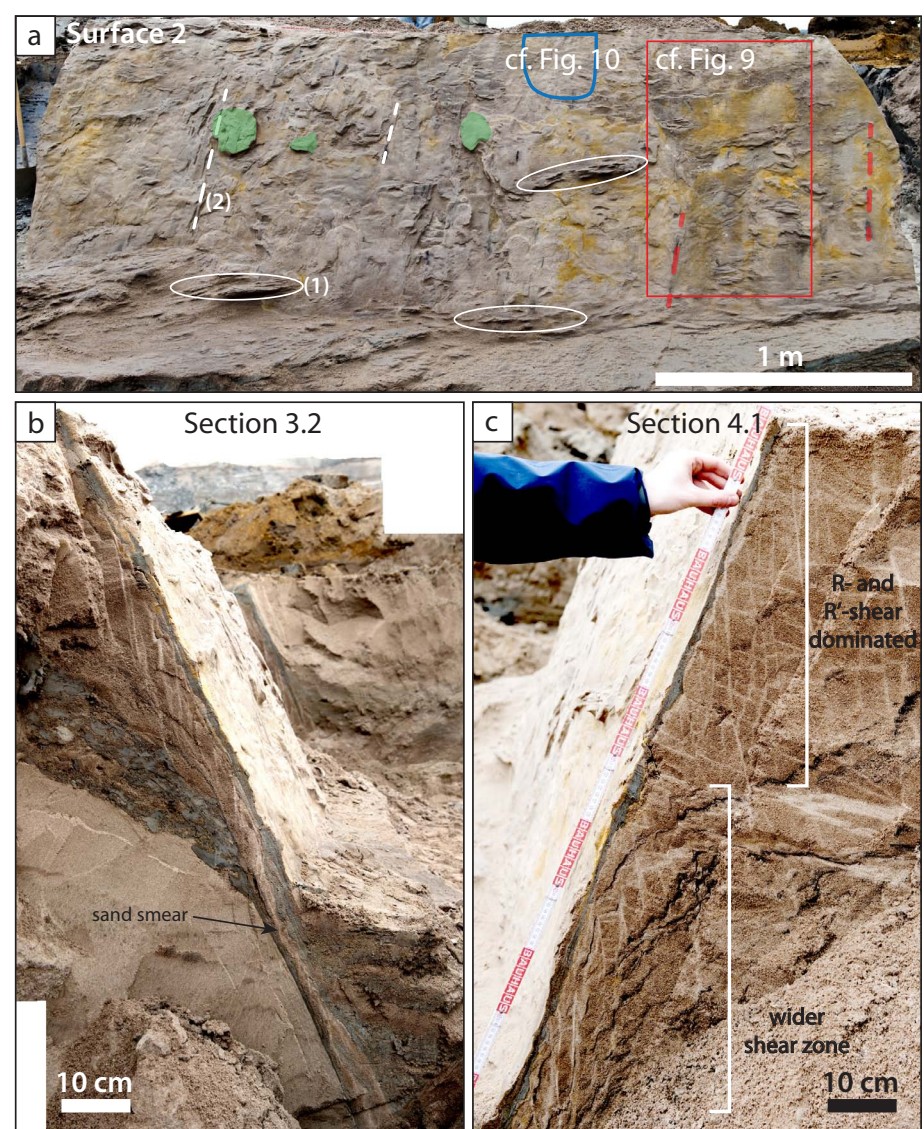

**Figure 7.** (a) Excavated surface 2 and corresponding vertical profiles 3.2 at the NW side (b) and 4.1 at the SE side (c). (b) A continuous sand smear developed between two continuous clay smears. (c) Intense footwall deformation is reflected in stair-stepping clay smear and strong variations in thickness.



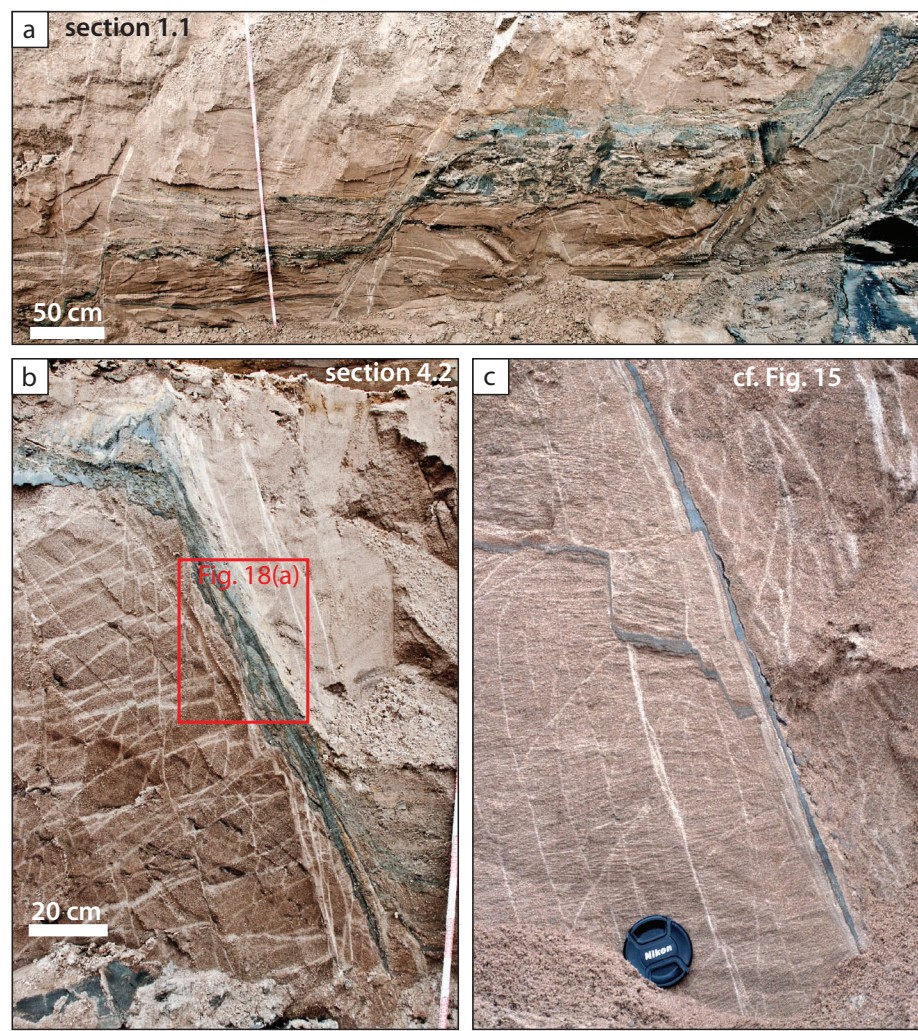

**Figure 8.** Vertical sections 1.1 (a) and 4.2. (b). (c) vertical section 5. This section was excavated during a different field campaign, but at the same fault and elevation at the talus next to the excavation site.





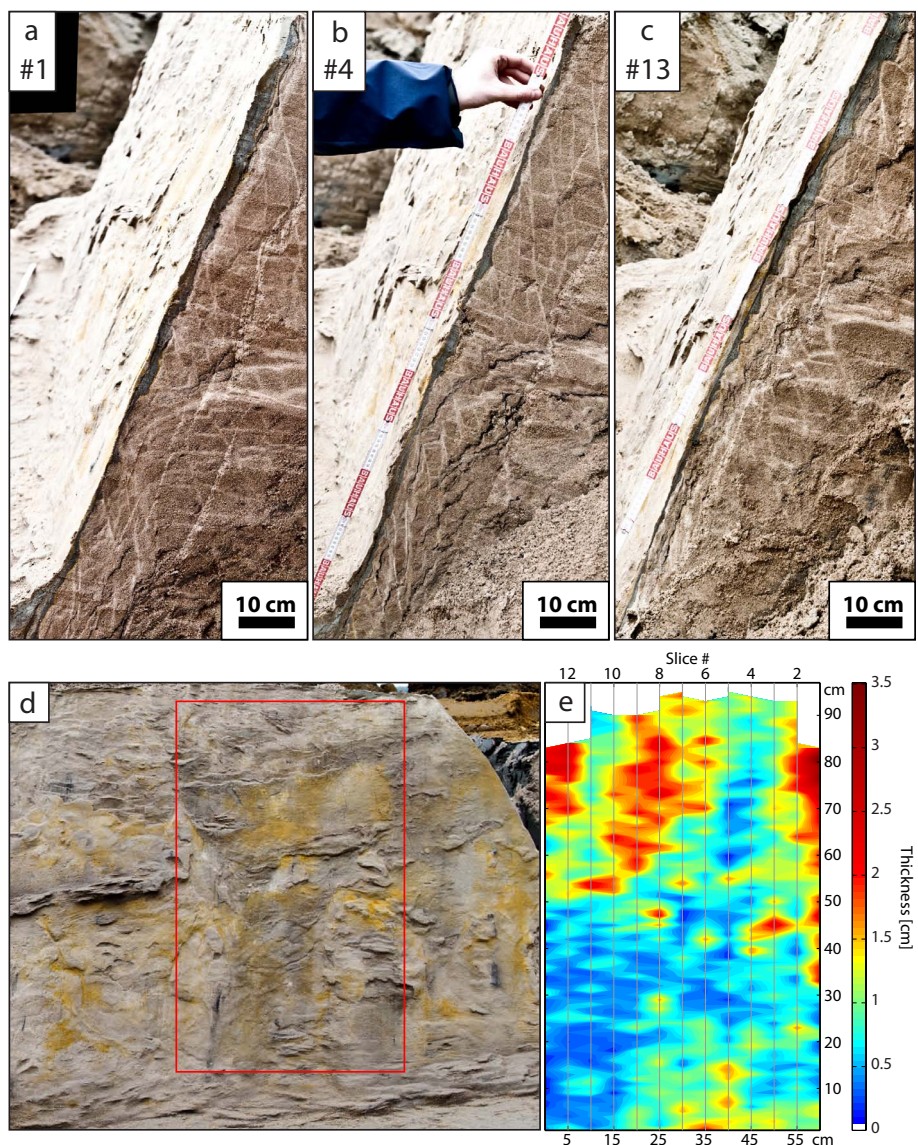

**Figure 9.** A thickness map of a section of surface 2 compiled from 13 vertical profiles (e.g. a – c). (d) Location of the thickness map. (e) Color coded thickness map. Thickness varies vertically and laterally. A trend towards thinner clay smears appears at the lower left.




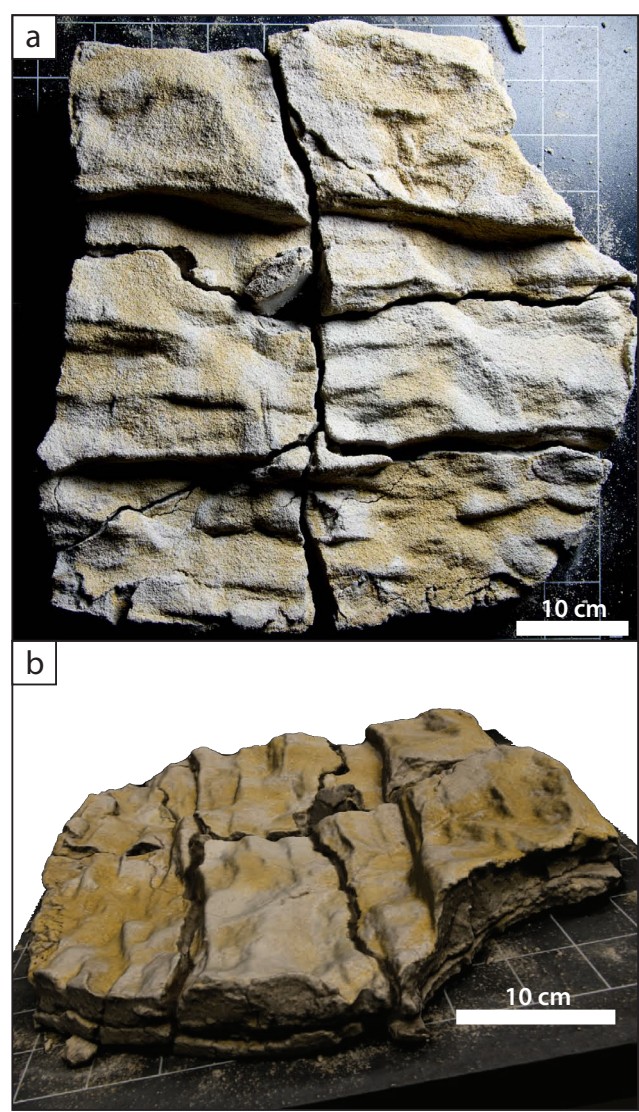

**Figure 10.** Geometry of stair-steps at the footwall side of a clay smear (cf. Fig. 7(a) for location). (a) Shadows reveal the structure. Light source in upper right corner. (b) Oblique view at a 3D model created by photogrammetry. Stair-steps resemble sudden changes in clay smear thickness. 3D model is available as online supplement.




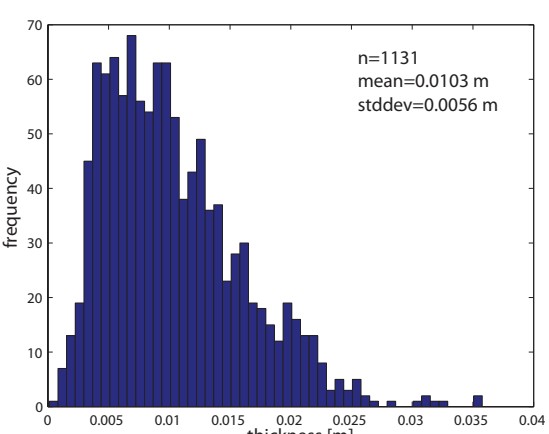

**Figure 11.** Histogram of 3D clay smear thickness shows log-normal distribution.





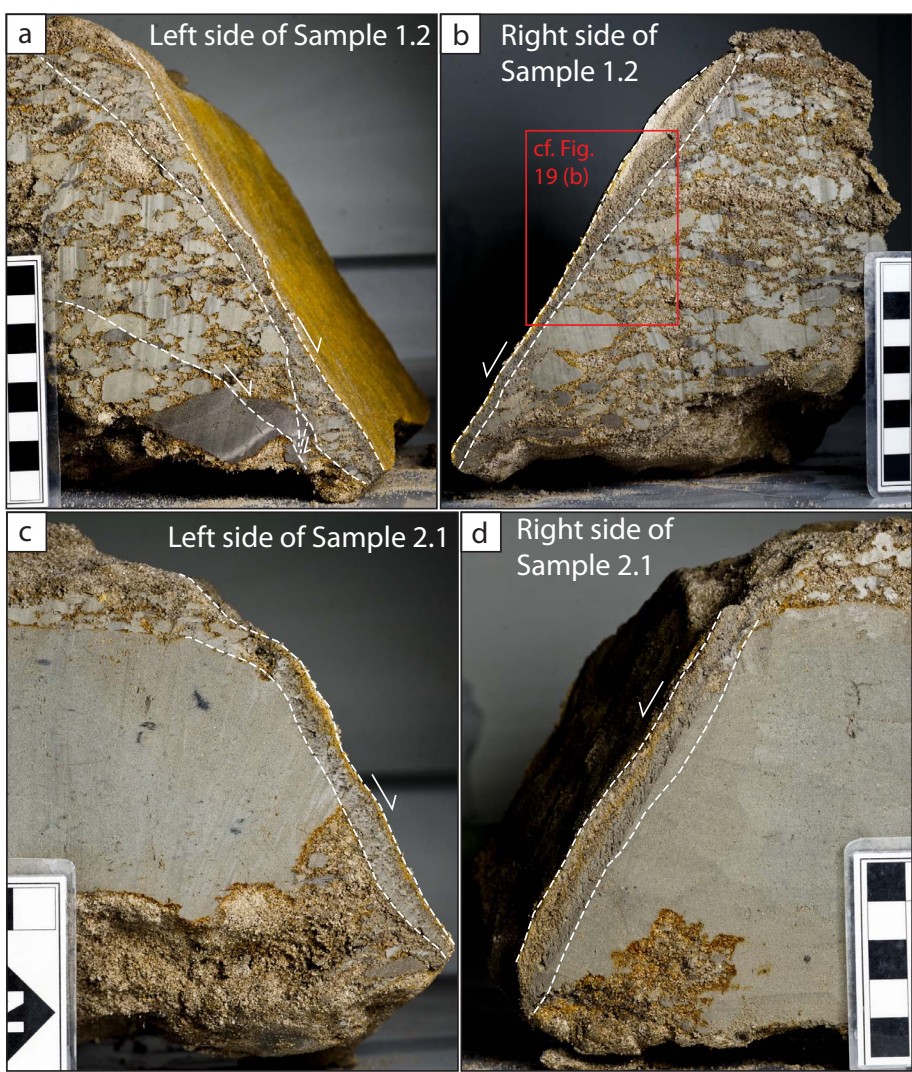

**Figure 12.** Cleaned samples extracted from profiles 1.2 & 2.1. Each sample has a width of about 20 cm. (a) & (b) show both sides of sample 1.2; (c) & (d) show both sides of sample 2.1. Dashed lines indicate the shear zone boundaries. Note the increasing amount of grain-scale mixing with greater distance of the source clay. Red rectangle marks the detail shown in Fig. 18(b).

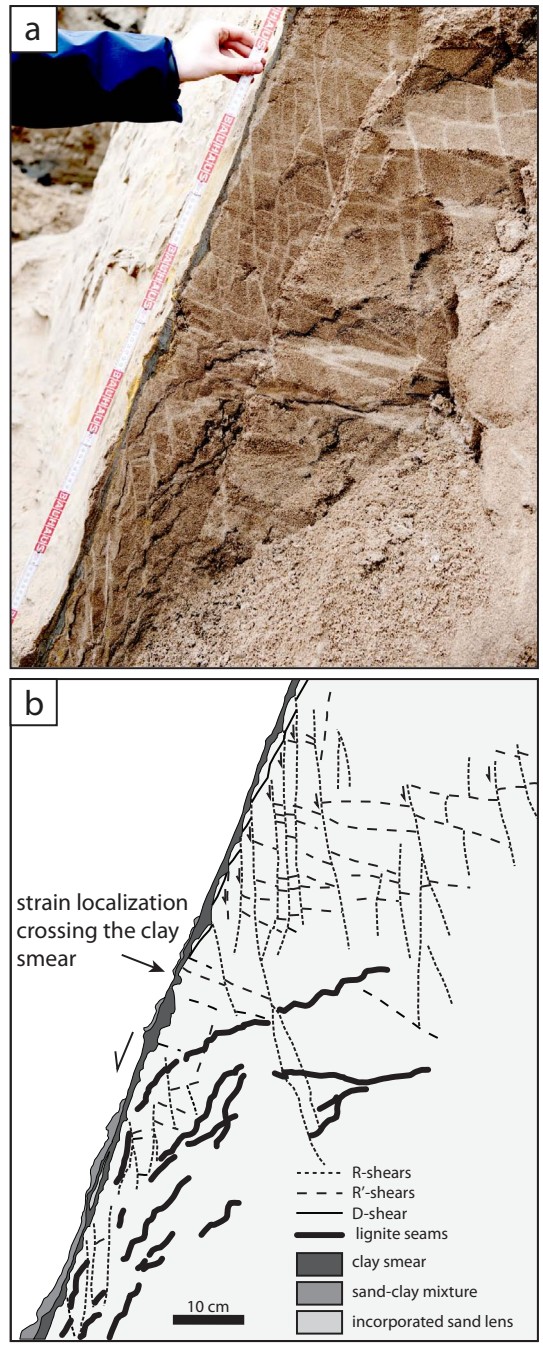

**Figure 13.** R-shears in the footwall deform the clay smear in a stair-stepping pattern. Clay smear is thinnest in elongation of the R-shears. Intense grain-scale mixing at the hanging-wall side occurs in the lower half of the section, coinciding with a D-shear crossing the clay smear approximately at the position marked by an arrow.



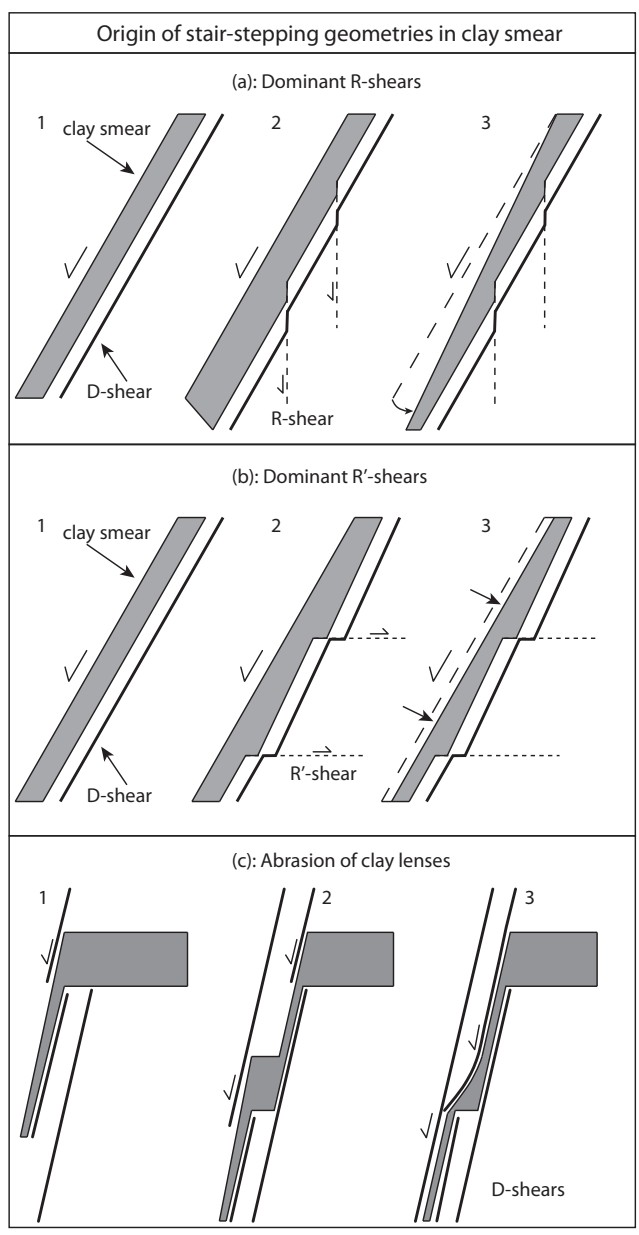

**Figure 14.** Possible origin of stair-stepping geometries in clay smear. (a) Dominant R-shears offset the sand-clay interface and D- and R'-shears (cf. Fig. 13). Plastic deformation of the clay smear is required to preserve the clay volume. (b) Dominant R'-shears offset the clay-sand interface and D- and R-shears (cf. Fig. 15). Plastic deformation of the clay is required to preserve the clay volume. (c) Clay lenses whose hanging-wall side is abraded can form stair-steps in the footwall in the absence of R- or R'-shears. Clay is not required to behave plastic.





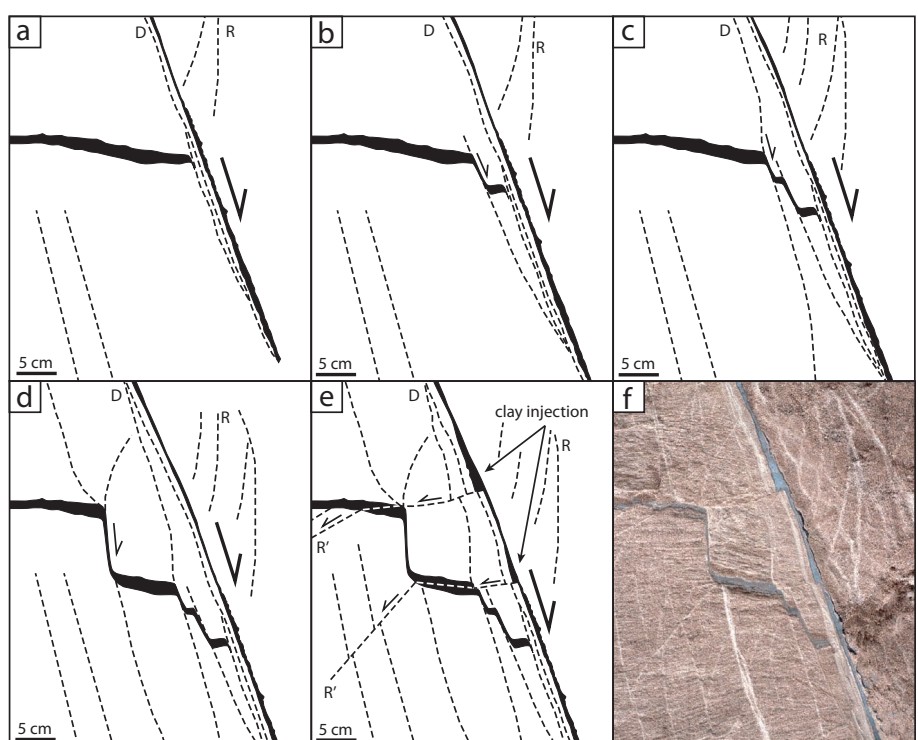

**Figure 15.** Sketch illustrating the evolution of a clay smear with stair-stepping due to R'-Shears. R'-shears are refracted to a very shallow dip within clay layers and offsets these.





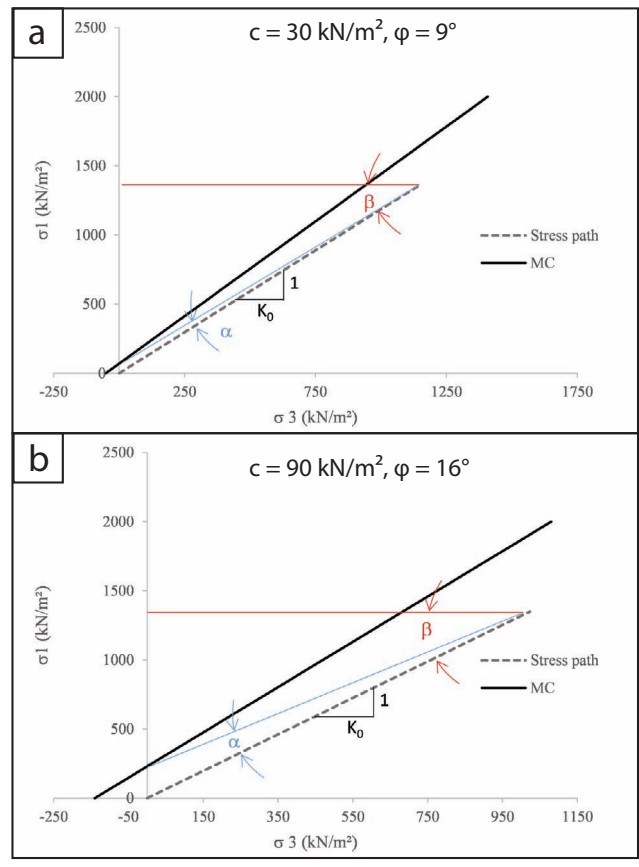

**Figure 16.** (a) Estimation of $\beta$ and $\alpha$ for c = 30 kN/m$^2$ and $\phi$ = 9°. (b) Estimation of $\beta$ and $\alpha$ for c = 90 kN/m$^2$ and $\phi$ = 16°.

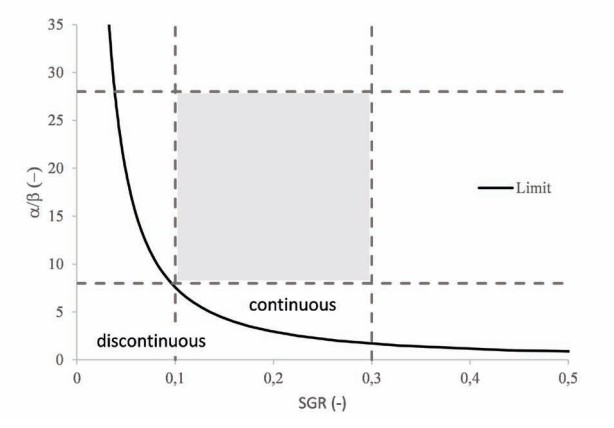

**Figure 17.** Evaluation of the continuity of the clay smear after Kleine Vennekate (2013).



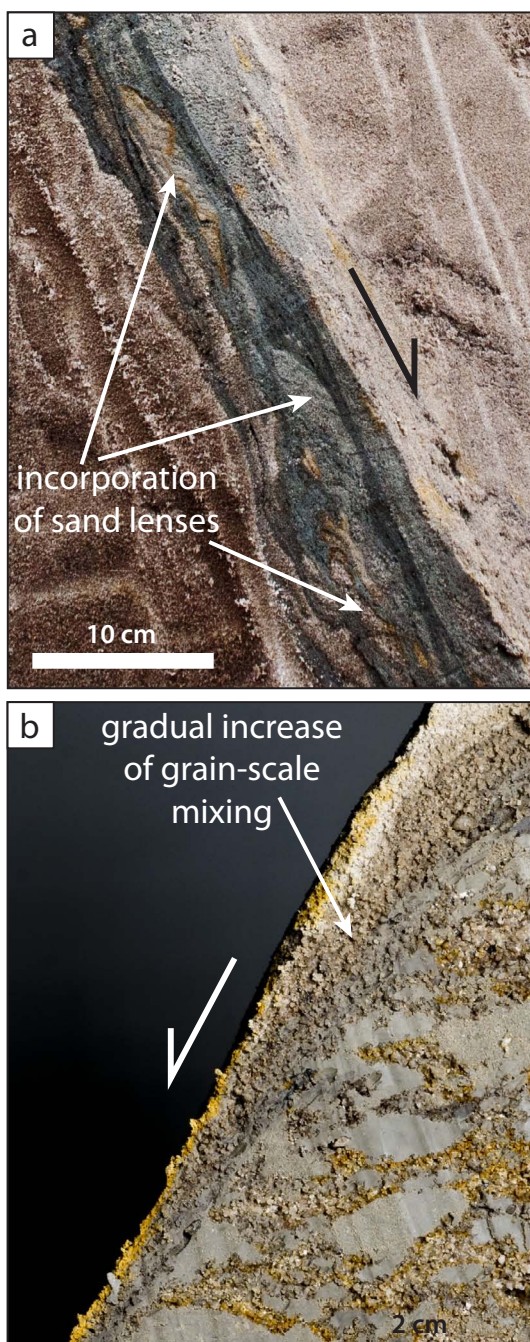

**Figure 18.** (a) Sand lenses are incorporated into a clay smear by grain-scale mixing and lead to increasing thickness of the clay rich fault material. (b) A layer of loosely packed rip-up clasts embedded in sand forms a continuous clay smear by grain-scale mixing. Due to the close spacing of sand and clay grains are mixed immediately during shearing. Note how the sand content increases with increasing distance to the source layer.



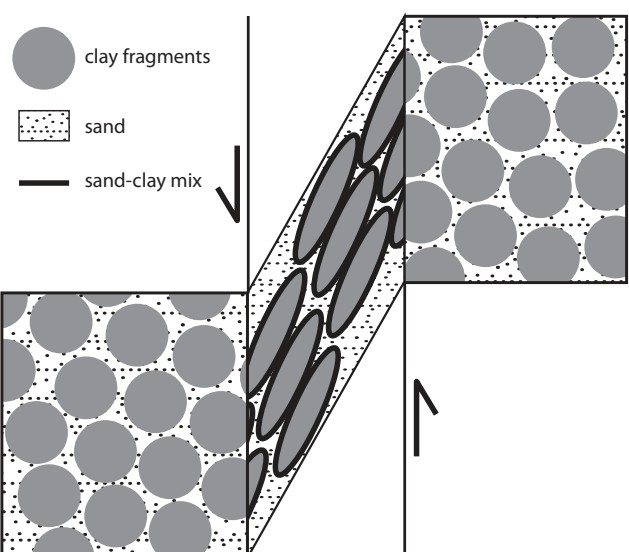

**Figure 19.** Concept of the simulation model testing the effect of mixing rate and clay fragment size on grain-scale mixing. Clay fragments embedded in a sand matrix are subject to simple shear. Sand-clay mixing zone around clay fragments increases with strain.





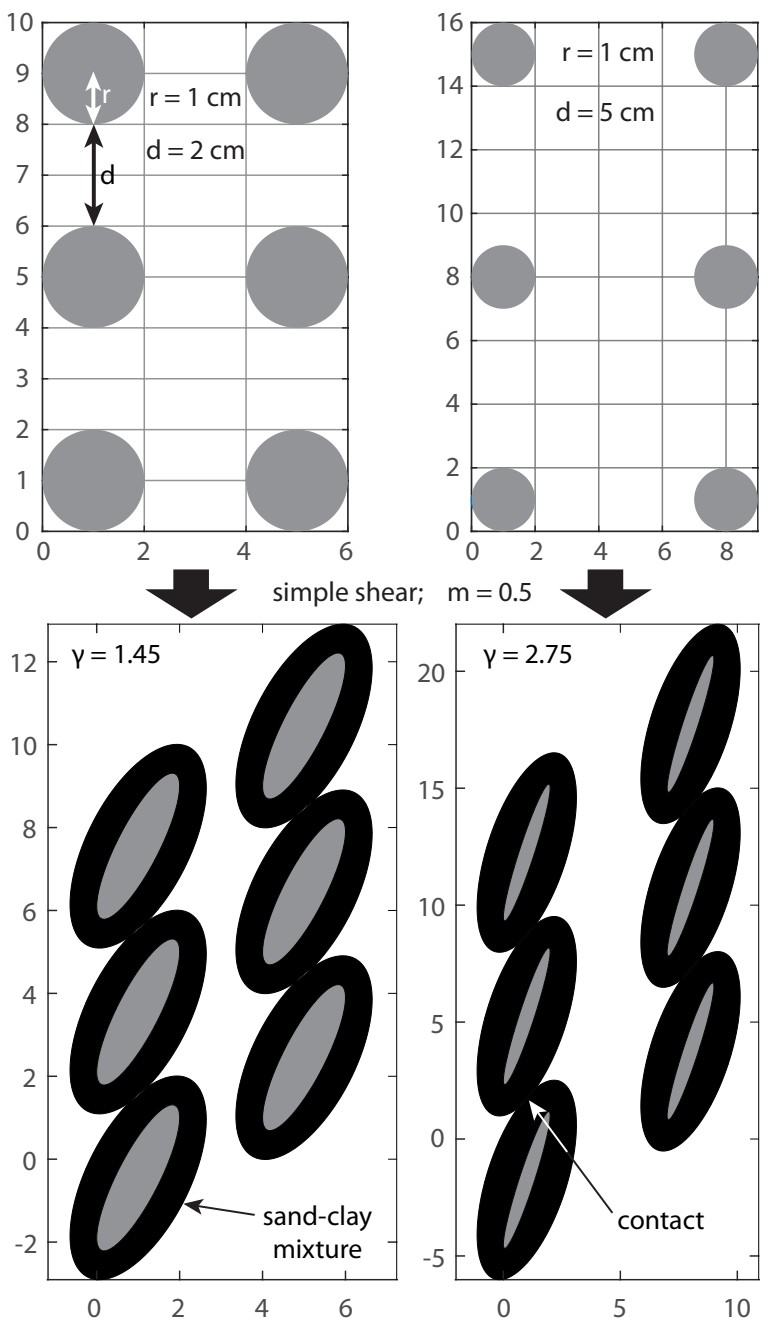

**Figure 20.** Example results of the mixing simulations for clay fragment with 1 cm radius and distances between fragments of 2 and 5 cm. Strain was increased until sand-clay mixtures surrounding the clay fragments touch. Initial packings are shown as well as the sheared models at maximum strain. Larger distances between fragments require higher strain at the same rate of mixing to connect via mixing.





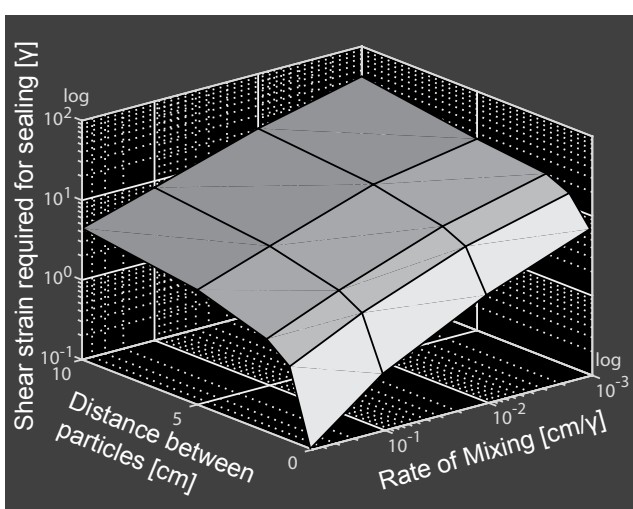

**Figure 21.** 3D plot relating rate of mixing and distance between clay fragments to strain required for fragments to connect via mixing. With a mixing rate of 0 sheared fragments will never touch each other.



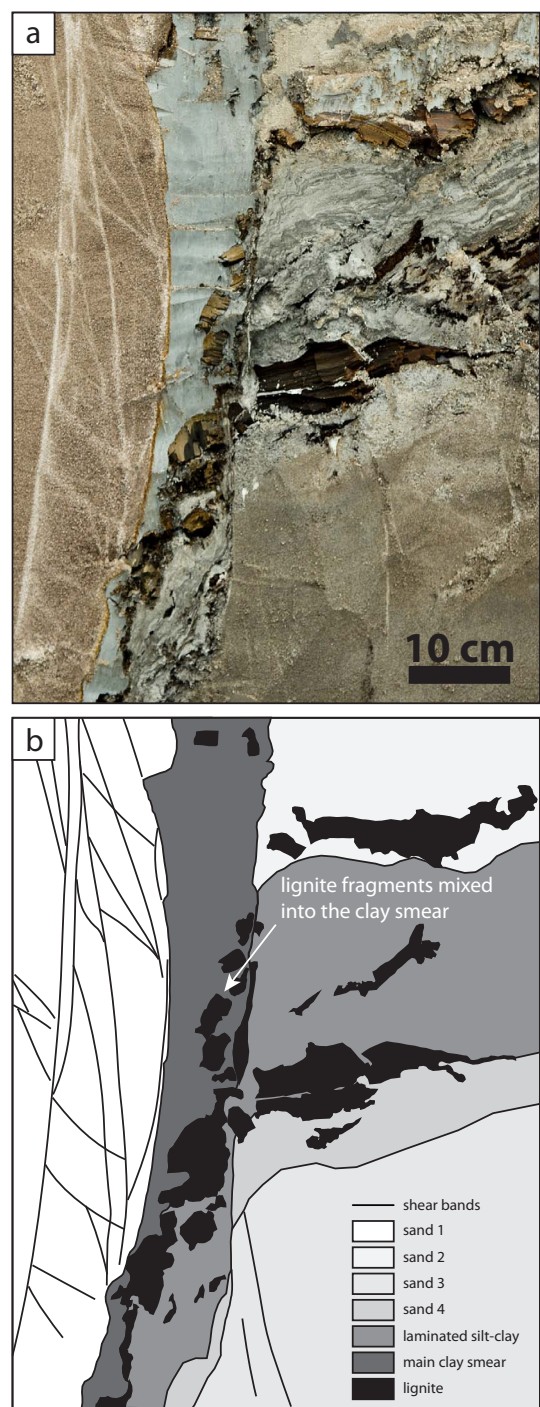

**Figure 22.** (a) Vertical cross-section on the fifth floor of the Hambach mine documented during a different field campaign. Fault, rough location and elevation correspond to outcrops in this paper. (b) Interpretation showing brittle lignite fragments transported into clay smear. Fragments appear aligned within clay smear.





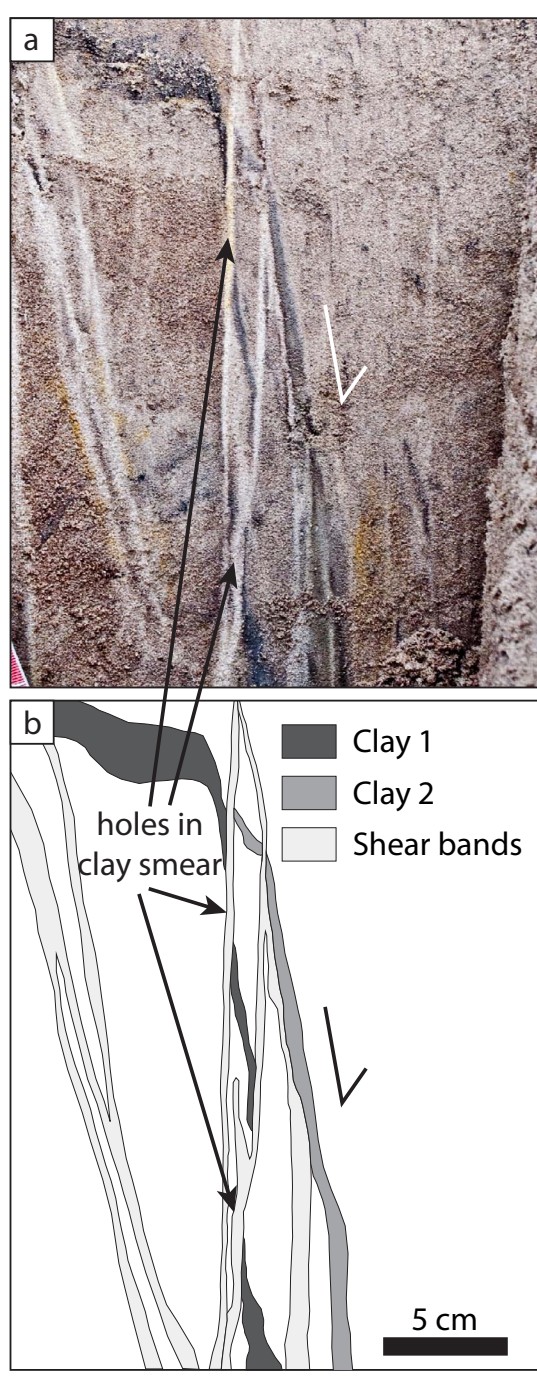

**Figure 23.** Hole in clay smear in a vertical section due to crosscutting shear bands. (a) Field photograph. (b) interpretation of shear bands and clays.



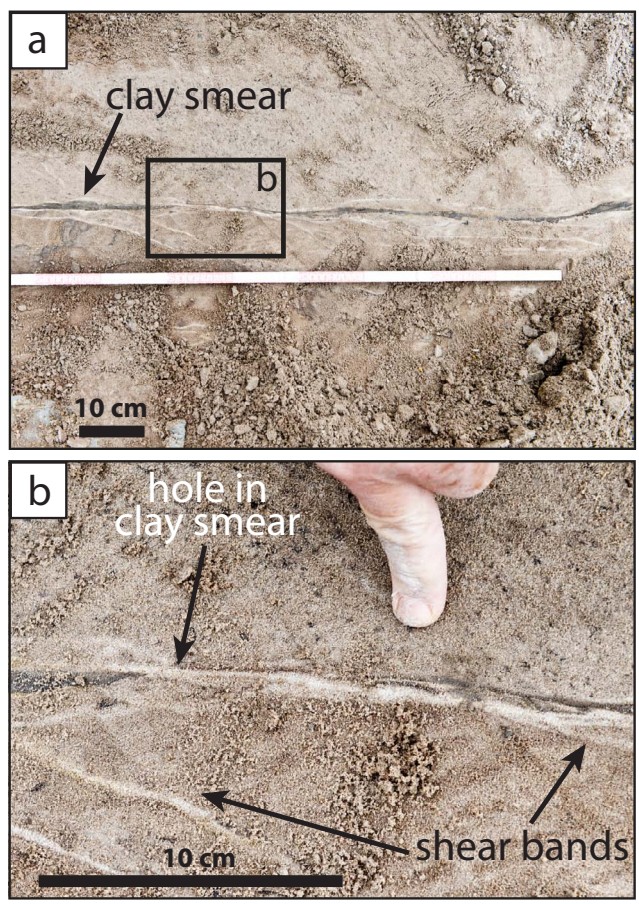

**Figure 24.** Hole in a clay smear in a horizontal profile due to crosscutting shear bands. (a) Overview photograph. (b) Detail of (a) showing the hole at the crosscutting shear band.