# Peer review of "Mechanisms of clay smear formation in unconsolidated sediments insights from 3D observations of excavated normal faults"

_Solid Earth, 2016_

## Referee Comment (RC1) · Anonymous Referee #1 · 7 Feb 2016

The manuscript investigates the mechanisms of clay smear using an exceptional outcrop exposure created by the Authors by excavating around a normal fault in a lignite mine in Germany. The Authors integrate the detailed field observations with: a) a 3D model created by photogrammetry that is used to map the 3D clay smear thickness and b) a model to characterize the effect of clay fragment size and rate of mixing on the evolution of sand-clay gouge.

I think that the manuscript presents a very detailed work, based on a unique 3D fault exposure, with fundamental observations for the understanding of the 3D evolution of clay smear. Therefore I strongly support the publication of this manuscript in Solid Earth.
[Figure]

In reading the manuscript, with its length, the 24 figures, the Appendix, the 3D model to be viewed with Matlab, the Matlab code in supplement material to evaluate the evolution of clay-sand gouge, I was wondering if it would be better to split this huge work in at least two manuscripts: a) one dealing with geometrical characterization of fault and clay smears; b) the other dealing with 3D models, detailed analytical outcomes (for example at lines 289-290 the sentence "The measured thickness data show a log-normal distribution" and the associated figure seems to be not well explained), and the modeling part.

In same parts of the manuscript I have not been able to see in the figures, what it is mentioned in the text or the figures deserve a better labeling. For example, lines 195-195 mentioning Figure 5: I am not able to see both hanging-wall and footwall cut off (labeling the cut-off would help the reader). The text at lines 229-231 is not clear or in other words the figures are not clearly explained by the text. It would be helpful to label R R1 and D-shears in figure 7 since it is the first time this terminology, together with a fault image for it, is introduced in the manuscript. Can you label D-shear in figure 13: it took me a lot of time to pick-up the D-shear position.

Paragraph 5.1 on the origin of stair-stepping geometries. Some jumps forward back forward (Figure 13-14-15) in mentioning figures and model have created a bit of confusion during my reading. I suggest first describing the observations and then presenting the model.

Lines 482-483. In the model there is the assumption of circular clay fragments. Since clay minerals are platy minerals I suggest to better justify this assumption.
* * *

---

## Referee Comment (RC2) · G. Yielding (Referee) · 22 Feb 2016

This manuscript provides an excellent documentation of the 3D structure of a meso-scale clay smear in an unconsolidated sand-clay sequence. There has long been a need for this type of study, which unfortunately is difficult to perform in practice. The authors have carefully chosen a good field opportunity, with industrial back-up for its execution. I have no hesitation in strongly recommending the manuscript for publication.

My personal interest in this work comes from its applicability in subsurface data, for example fault seal analysis in hydrocarbon exploration/production and CCS. In this regard, I think a very useful addition to the figures would be explicit SGR results for each

section where FW and HW parts of the clay are visible. SGR values are mentioned in places in the text but in a rather approximate way.

On page 5, line 16, the comment should be more carefully worded to avoid mis-interpretation - I suggest "If we are looking for faults with SGR<0.2, single source clays have to be <20cm thick if the fault throw = 1.0m."

The discussion of Figure 11 (clay smear thickness histogram) should include a con-sideration of sampling artifacts at the small-thickness end of the distribution. This is analogous to the concerns about fault-population sampling (e.g. Pickering et al 1995) where truncation at small sizes distorts the statistical fit. I would also suggest plotting the fitted log-normal distribution onto the histogram.

In general I feel that it is good to have the discussion/modelling section here in this same paper as the outcrop observations, in contrast to Anonymous Referee #1. How-ever, the Matlab model presented on p.13 (lines 6-18) and Figures 19-21 does not seem particularly insightful, so maybe it could be omitted to shorten the paper a little, or moved to a second Appendix.

Some minor technical corrections are as follows:

p.1, line 12: sheared not shared

p.2, line 29, insert 'and' after 'faults,'

p.3, line 19, 'in relays' not 'of relays'

p.6, lines 14-16 would be better moved to around line 3, as they are general observa-tions

p.7, line 21: R- and R'- shears are absent... there seem to be lots of them on the lower part of Figure 7c. And also, it does NOT seem that the shear zone is wider; refer to interpretation in Fig.13b.

p.7, lines 27-28: this sentence refers to Fig.22 and is out of sequence.

p.8, line 8: omit 'with'

p.8, line 10: 'sections 3, 9, 10'....section 9 does NOT show thin smear at the footwall cutoff: perhaps 3, 10 & 11?

p.8, line 19: cast not casted

p.8, line 25-26: I cannot understand this sentence: what is 'footwall cutoff on the hanging-wall side'?

p.10, line 2: principal not principle

p.10, line 18: given that the Kleine Vennekate reference is relatively inaccessible, I suggest that this b/a vs SGR plot be explained more, perhaps in the caption to Fig.17.

p.13, line 4: insert 'implies' after 'smear'

p.15, line 2: than not as

p.15, lines 7-8: meaning unclear, please re-write

Figure 6a, very pretty, but please state in the caption that the colours have been added to improve clarity!

Figure 9: make part (e) exactly the same size as the red box in part (d), to improve the reader's ability to match up features in the two images

Figure 13: in agreement with Referee #1, improve annotation of D-shear

---

## Author Comment (AC1) · 15 Apr 2016

Answer to the review of Graham Yielding:

Dear Graham Yielding,

We are very grateful for your kind and helpful review and the recommendation for publication. In the following we answer to the individual issues that were pointed out to increase the quality of the manuscript.

Kind regards,

Michael Kettermann

1. *In this regard, I think a very useful addition to the figures would be explicit SGR results for each section where FW and HW parts of the clay are visible. SGR values are mentioned in places in the text but in a rather approximate way.*

→**We agree and add a table summarizing known and estimated SGR value for all cross-sections.**

| Section | Σ Clay thickness [cm] | Throw [cm] | SGR |
|---------|----------------------|------------|-----|
| 1.1 | ~15(left)/25(right) | 48(left)/54(right) | 0.3(left)/0.46(right)/0.2(total) |
| 1.2 | 17 | 54 | 0.3 |
| 2.1 | 19 | 57 | 0.3 |
| 2.2 | - | - | - |
| 3.1 | - | - | - |
| 3.2 | 30* | 70 | 0.4 |
| 4.1 | 6 | 120* | 0.05 |
| 4.2 | 20 | 90 | 0.2 |

**Table 1: Summary of SGR values in the presented cross-sections. In section 1.1 two fault strands are visible (cf. Fig. 8a), source clay thickness and throw values are given separately and SGR values provided for each fault strand and for the total throw. Estimated values marked with *.**

2. *"On page 5, line 16, the comment should be more carefully worded to avoid misinterpretation - I suggest "If we are looking for faults with SGR<0.2, single source clays have to be <20cm thick if the fault throw = 1.0m.""*

→**Agreed and changed accordingly.**

3. *The discussion of Figure 11 (clay smear thickness histogram) should include a consideration of sampling artifacts at the small-thickness end of the distribution. This is analogous to the concerns about fault-population sampling (e.g. Pickering et al 1995) where truncation at small sizes distorts the statistical fit. I would also suggest plotting the fitted log-normal distribution onto the histogram.*

→ **This is a good point. We modified the histogram to a more useful bin-size of 1 mm and added the fitted log-normal distribution. We added the following text:**

*"One has to consider, though, that a sampling bias at the lower end of the range might affect the distribution (e.g. Pickering et al., 1995). Thinner clay smear or holes may appear between sampling points and are then not resolved. However, visual inspection showed no holes in this part of the smear."*

[Figure]

4. *In general I feel that it is good to have the discussion/modelling section here in this same paper as the outcrop observations, in contrast to Anonymous Referee #1.*
   → **We think so as well and keep the manuscript as one work.**

5. *However, the Matlab model presented on p.13 (lines 6-18) and Figures 19-21 does not seem particularly insightful, so maybe it could be omitted to shorten the paper a little, or moved to a second Appendix.*
   →**We agree that this model mainly points out some of the relevant parameters for mixing, which are admittedly quite intuitive and this doesn't justify the length of the paragraph with three figures. However, we see a value in pointing this out to the reader as some kind of invitation to further investigate the processes of grain-scale mixing. The details of the model and the figures are not required in the text though and we moved most of it to Appendix B to shorten the main text distinctly.**
   **The shortened paragraph now reads:**
   *"To explore the effect of clay fragment size and rate of mixing on the evolution of sand-clay gouge, we designed a simple simulation (Matlab, 2015; code in online supplement) where circular clay fragments in a sand matrix are subject to homogeneous simple shear. A detailed description and figures of this model can be found in Appendix B. The results show a logarithmic relation between rate of mixing, distance between particles and the strain required to produce an effective seal by mixing. The initial packing will have an influence on the required strain as well as the distribution of mixing (e.g. stronger mixing at the top of clay fragments), however this will be subject of further research."*

   **Appendix B now has the three Figures 20, 21 and 22 (now B1, B2 and B3) attached and reads:**

*"The general idea of this model is to investigate parameters controlling grain-scale mixing in clay sand sequences. As a basic geometry we chose a number of circular clay fragments with dimensions in the order of mm to cm as observed e.g. in Sample 1.2 (c.f. Fig. 12a) which are embedded in a sand matrix (Fig. B1). This sediment package is then faulted with a certain shear-band width using a simple simulation (Matlab, 2015; code in online supplement). With increasing shear strain a sand-clay mixed seam around the fragments develops and increases in thickness. We ran five series of simulations with initially circular objects representing clay fragments. The rate of mixing is defined as $m = \Delta T/\Delta \gamma$, where $\Delta T$ is the change in thickness of the mixed seam per unit shear strain and $\Delta \gamma$ is the change in shear strain. The thickness of the mixing seam at a given shear strain is then $T = \gamma * m$. Simple shear is then applied to the model and shear strain is increased in steps of 0.05. This was done for five distances between clay fragments (0.1, 1, 2, 5 and 10 cm radius) and four rates of mixing (0.001, 0.01, 0.1 and 0.5). Using the 'intersections' algorithm (Schwarz, 2010) the code finds the strain at which the ellipses intersect (i.e. clay fragments touch). While a mixing rate of 0.5 is certainly unrealistically high, it serves well for illustrating the procedure (Fig. B2). The results show a logarithmic relation between rate of mixing, distance between particles and the strain required to produce an effective seal by mixing (Fig. B3)."*

**Minor technical corrections:**

6. *p.1, line 12: sheared not shared* → **changed accordingly**

7. *p.2, line 29, insert 'and' after 'faults,'* → **changed accordingly**

8. *p.3, line 19, 'in relays' not 'of relays'* → **changed accordingly**

9. *p.6, lines 14-16 would be better moved to around line 3, as they are general observations*
    → **Agreed and changed accordingly. Lines 14-16 removed and line 3 ff. now reads: "At first look, both clay smear surfaces contain many sub-horizontal clay ledges with fine horizontal layering locally visible (ellipses labeled (1) in Fig. 5a and Fig. 7a). The clay smear surface is colored by yellowish iron hydroxides. Black striations (dashed lines labeled (2) in Fig. 5a and Fig. 7a) are interpreted to be the result of the fault moving past lignite fragments and they can deviate up to 10° from dip-slip. Vertical sections at both sides of the clay smear are shown in Figure 5b & c and 7b & c."**

10. *p.7, line 21: R- and R'- shears are absent... there seem to be lots of them on the lower part of Figure 7c. And also, it does NOT seem that the shear zone is wider; refer to interpretation in Fig.13b.*
    → **This is true. We changed the text to: "In the lower part of this section we observe less R'-shears and the D-shear described in the upper part is absent. R-shears and diffuse deformation indicated by sheared lignite seams imply a narrower shear zone compared to the upper part of Figure 7c."**

**Additionally, we modified Figure 7c indicating our interpretation of the shear zone widths and following the suggestions of Reviewer #1 we exemplary indicated R-, R' and D-shears.**

[Figure]

11. *p.7, lines 27-28: this sentence refers to Fig.22 and is out of sequence.*

→ **This sentence is intended to present the data from an additional cross-section (from a different field campaign) to the reader and should be in this paragraph for reasons of completeness. However, to improve the readability and clarity we rephrase the paragraph to:** *"Two additional cross-sections are shown in this article from different field campaigns in the same mine and at the same fault and level: (1) Cross-section 5 in Figure 8c (see Fig. 15 for interpretation) shows stair-stepping structures at the footwall side of the clay smear and numerous R-, R'- and D-shears forming two deformed clay smears. (2) We observed a thicker clay smear (~10 cm thickness) with brittle clay fragments entrained into the smear in a cross-section during another field work in 2015 as shown and interpreted in Figure 25a & b."*

12. *p.8, line 8: omit 'with'* → **changed accordingly**

13. *p.8, line 10: 'sections 3, 9, 10'....section 9 does NOT show thin smear at the footwall cutoff: perhaps 3, 10 & 11?*

→ **Yes, you are right, it should be sections 3, 10 and 11. Changed accordingly.**

14. *p.8, line 19: cast not casted* → **changed accordingly**

15. *p.8, line 25-26: I cannot understand this sentence: what is 'footwall cutoff on the hanging-wall side'?*

→ **We agree that this sentence is confusing. While footwall cutoff should be clear, the hanging-wall side refers to the "upper"-side of the clay smear, i.e. the side of the clay smear**

**facing the hanging-wall. We rephrase this sentence to clarify:** *"At the top of the sample, i.e. the hanging-wall side of the clay smear located at the footwall cutoff, we note the highest sand content that decreases further towards the footwall"*

16. *p.10, line 2: principal not principle* → **changed accordingly**

17. *p.10, line 18: given that the Kleine Vennekate reference is relatively inaccessible, I suggest that this b/a vs SGR plot be explained more, perhaps in the caption to Fig.17.*

→**Yes, this work is unfortunately not yet published in well accessible journal. We extended the description of the method and added a new graphic to aid the understanding. It now reads (Figure numbers don't match the manuscript):**

*"As stated before there are several empirical methods to predict the sealing potential of clay smears which are based on their actual deformed configuration. These methods are often criticized for overlooking the mechanical and hydraulic behavior of the sealing material. Based on numerical simulations, Kleine Vennekate (2013) proposed a new methodology to evaluate the continuity of the clay smear in a normal fault. This methodology takes into account not only the actual geometry of the deformed clay but also considers the stress state and the shear strength of the low permeable layer.*

*The evaluation of the stress state and shear strength follows the idea of the MCIP proposed by van der Zee (2003), which infers if the clay deformation occurred under a tension or compression regime for the lowest principal stress ($\sigma_3$). The method of Kleine Vennekate (2013) assesses two angles $\alpha$ and $\beta$ in a principal stress $\sigma_1$ and $\sigma_3$ diagram shown in Figure 1. Both angles relate the position of the stress state prior deformation to the Mohr Coulomb failure criterion. The angle $\alpha$ links them with the first principal stress axis ($\sigma_3 = 0$) whereas $\beta$ relates the stress state and the Mohr Coulomb criterion with the estimated stress path during deformation (horizontal line). A ratio $\frac{\beta}{\alpha} < 1$ implies that $\sigma_3$ will be negative during the deformation, otherwise $\sigma_3$ will be positive. Figure 1 shows an example of two points with different stress states and different $\frac{\beta}{\alpha}$ ratios. The deformed configuration is considered by using the shale gouge ratio. Both criteria, $\frac{\beta}{\alpha}$ and SGR, are then plotted together with a curve that marks the limit between a continuous and discontinuous clay gouge.*

*This methodology was followed to assess the continuity of the clay gouge in the excavated fault. Figure 2 and 3 show in a principal stress diagram ($\sigma_1$, $\sigma_3$) the Mohr-Coulomb limit state line and the assumed stress path together with the angles $\beta$ and $\alpha$ for c=30kN/m² $\varphi$ = 9° and c=90 kN/m² $\varphi$ = 16° respectively. The ratio $\frac{\beta}{\alpha}$ found can vary from a value of 8 up to a value of 28, implying that the value of $\sigma_3$ was positive during the deformation.*

*The continuity of the clay smear is then evaluated in figure 4. Here the limit between continuous and discontinuous clay smear is presented with the continuous line and the calculated upper and lower limit of both SGR and $\frac{\beta}{\alpha}$ are plotted with dashed lines. The shaded area represents all the possible combinations of SGR and $\frac{\beta}{\alpha}$. According to Kleine Vennkate (2013) it can be expected that the clay*

*smear would be continuous since this area is above the limit curve, the continuous clay smear zone, which is in agreement with the field observations."*

[Figure]

*Figure 1: Definition of parameters $\alpha$ and $\beta$ and estimation of $\frac{\beta}{\alpha}$ ratios for different stress states. Point 1 (p1) has a ratio $\frac{\beta}{\alpha} < 1$ meaning $\sigma_3$ is negative, whereas Point 2 has a ratio of $\frac{\beta}{\alpha} > 1$, meaning positive $\sigma_3$. Red line: Mohr-Coulomb failure criterion, blue dashed lines: stress-path during faulting, green dashed lines: connects beginning of the stress-path during faulting (i.e. p1 or p2) with the intersection of the failure criterion and the $\sigma_1$ axis.*

[Figure]

*Figure 2: Estimation of $\beta$ and $\alpha$ for c=30kN/m² and $\varphi$ = 9°. Ratio of $\frac{\beta}{\alpha} > 1$, meaning positive $\sigma_3$.*

[Figure]

*Figure 3: Estimation of β and α for c=90 kN/m² φ = 16. Ratio of $\frac{\beta}{\alpha} > 1$, meaning positive $\sigma_3$.*

[Figure]

*Figure 4: Evaluation of the continuity of the clay smear after Kleine Vennekate (2013). Shaded area represents all possible combinations of SGR and $\frac{\beta}{\alpha}$, for the presented data.*

18. *p.13, line 4: insert 'implies' after 'smear'* → **changed accordingly**

19. *p.15, line 2: than not as* → **changed accordingly**

20. *p.15, lines 7-8: meaning unclear, please re-write*

→ **Yes, something went wrong there. We rephrased to *"We report observations for faults in this study that are one to two orders of magnitude smaller than those described by Eichhubl et al. (2005), Faerseth (2006) or Aydin and Eyal (2002) but share similar characteristic structures. Thus we hypothesize that detailed observations on small scale faults can be transferred to faults at least one order of magnitude larger."***

21. *Figure 6a, very pretty, but please state in the caption that the colors have been added to improve clarity!*

→ **The caption now reads: *"(a) Detail of the SE side of surface 1 showing multiple thin clay veneers composing the bulk clay smear. Colors added manually to distinguish different clay layers."***

22. *Figure 9: make part (e) exactly the same size as the red box in part (d), to improve the reader's ability to match up features in the two images*

→ **Agreed and done.**

23. *Figure 13: in agreement with Referee #1, improve annotation of D-shear*

→ **Additionally to the legend we added labels for R-, R'- and D-shears to the interpreted image.**

---

## Author Comment (AC2) · 15 Apr 2016

Answer to Anonymous Referee #1:

We thank the anonymous referee for the positive feedback and comments that help to improve the manuscript. In the following we answer to all comments and state corrections and changes we made to the text following the referee's suggestions.

Kind regards,

Michael Kettermann

1. *"In reading the manuscript, with its length, the 24 figures, the Appendix, the 3D model to be viewed with Matlab, the Matlab code in supplement material to evaluate the evolution of clay-sand gouge, I was wondering if it would be better to split this huge work in at least two manuscripts: a) one dealing with geometrical characterization of fault and clay smears; b) the other dealing with 3D models, detailed analytical outcomes (for example at lines 289-290 the sentence "The measured thickness data show a lognormal distribution" and the associated figure seems to be not well explained), and the modeling part."*
→ **We agree that the manuscript is quite long and presents different approaches and techniques. However, we think that it is important to keep the different sections as part of one coherent manuscript. The mentioned 3D thickness map and model for instance essentially need the description of the respective outcrop to be of value. In agreement with referee #2 we will keep the work as one manuscript.**

**In accordance with Referee #2 we also agree that the presented Matlab mixing model mainly points out some of the relevant parameters for mixing, which are admittedly quite intuitive and this doesn't justify the length of the paragraph with three figures. However, we see a value in pointing this out to the reader as some kind of invitation to further investigate the processes of grain-scale mixing. The details of the model and the figures are not required in the text, though, and we moved most of it to Appendix B to shorten the main text distinctly.**

**The shortened paragraph now reads:**
***"To explore the effect of clay fragment size and rate of mixing on the evolution of sand-clay gouge, we designed a simple simulation (Matlab, 2015; code in online supplement) where circular clay fragments in a sand matrix are subject to homogeneous simple shear. A detailed description and figures of this model can be found in Appendix B. The results show a logarithmic relation between rate of mixing, distance between particles and the strain required to produce an effective seal by mixing. The initial packing will have an influence on the required strain as well as the distribution of mixing (e.g. stronger mixing at the top of clay fragments), however this will be subject of further research."***

**Appendix B now has the three Figures 20, 21 and 22 (now B1, B2 and B3) attached and reads:**

*"The general idea of this model is to investigate parameters controlling grain-scale mixing in clay sand sequences. As a basic geometry we chose a number of circular clay fragments with dimensions in the order of mm to cm as observed e.g. in Sample 1.2 (c.f. Fig. 12a) which are embedded in a sand matrix (Fig. B1). This sediment package is then faulted with a certain shear-band width using a simple simulation (Matlab, 2015; code in online supplement). With increasing shear strain a sand-clay mixed seam around the fragments develops and increases in thickness. We ran five series of simulations with initially circular objects representing clay fragments. The rate of mixing is defined as $m = \Delta T/\Delta\gamma$, where $\Delta T$ is the change in thickness of the mixed seam per unit shear strain and $\Delta\gamma$ is the change in shear strain. The thickness of the mixing seam at a given shear strain is then $T = \gamma * m$. Simple shear is then applied to the model and shear strain is increased in steps of 0.05. This was done for five distances between clay fragments (0.1, 1, 2, 5 and 10 cm radius) and four rates of mixing (0.001, 0.01, 0.1 and 0.5). Using the 'intersections' algorithm (Schwarz, 2010) the code finds the strain at which the ellipses intersect (i.e. clay fragments touch). While a mixing rate of 0.5 is certainly unrealistically high, it serves well for illustrating the procedure (Fig. B2). The results show a logarithmic relation between rate of mixing, distance between particles and the strain required to produce an effective seal by mixing (Fig. B3)."*

2. *For example at lines 289-290 the sentence "The measured thickness data show a lognormal distribution" and the associated figure seems to be not well explained.*
   **→ We agree that this sentence lacks clarity and information and is somehow badly positioned in the text. We move the sentence backwards in the text and rephrased. The paragraph now reads:**
   *"The thickness map (Fig. 9e) shows that the clay smear is patchy, with a gradual change between profiles. A general trend is towards thinner clay in lower left, but horizontally elongated thicker patches are distributed over the entire area. Thick clay smear is located in the lower central part as well as the upper 50 cm of the smear. However, even close to the footwall cutoff of the source clay, thin clay smear (less than 1 cm) occurs in sections 3, 10 and 11. Plotting the measured thickness data in a histogram (1 mm bin size, N = 1131) results in a log-normal distribution similar to those shown by Navarro (2002) from 2D profiles. The histogram including a fitted log-normal distribution and all essential data is shown in Figure 11."*

3. *In some parts of the manuscript I have not been able to see in the figures, what it is mentioned in the text or the figures deserve a better labeling. For example, lines 195- 195 mentioning Figure 5: I am not able to see both hanging-wall and footwall cut off (labeling the cut-off would help the reader).*

   **→To clarify the structures mentioned in the text we added more labels to some figures as shown below:**

[Figure]

a footwall cutoff
clay
Surface 1
detail Fig. 6
hanging-wall cutoff
1 m
(1)
(2)

b Section 1.2
10 cm

c Section 2.1
Shear Zone Boundary
10 cm

a Surface 2
cf. Fig. 10
cf. Fig. 9
(2)
(1)
1 m

b Section 3.2
source clay
clay smear 1
clay smear 2
sand smear
10 cm

c Section 4.1
R- and R'-shear dominated
D-shear
shear zone width
shear zone width
narrower shear zone
10 cm

[Figure]

N = 1131
Min = 0.012 cm
Max = 3.6 cm
Mean = 1.03 cm
stddev= 0.56 cm
BinWidth = 0.1 cm
n_bins = 36

fitted log-normal distribution

frequency
thickness [cm]

a section 1.1
fault strand 2
fault strand 1
source clay
50 cm

b section 4.2
source clay
Fig. 18(a)
20 cm

c cf. Fig. 15
stair-steps

4. *The text at lines 229-231 is not clear or in other words the figures are not clearly explained by the text.*
   ➔ **Unfortunately your line numbering does not correspond to the line-numbering in the manuscript (1-~34 for each page) and hence it is not clear to us which sentences/figures you refer to here. If you clarify this, we are glad to improve the text.**

5. *It would be helpful to label R R1 and D-shears in figure 7 since it is the first time this terminology, together with a fault image for it, is introduced in the manuscript. Can you label D-shear in figure 13: it took me a lot of time to pick-up the D-shear position.*
   ➔**Yes, this would improve the clarity of the figures. We put labels for R-, R' and D-shears in both Figure 7c and Figure 13.**

[Figure]

6. *Paragraph 5.1 on the origin of stair-stepping geometries. Some jumps forward back forward (Figure 13-14-15) in mentioning figures and model have created a bit of confusion during my reading. I suggest first describing the observations and then presenting the model.*
→ **We agree and restructure this paragraph to:**

*"In the following we show observations of three different types of stair-stepping geometries in clay smears (related to R-shears, R'-shears or abrasion) and present models explaining their formation and possible influence on the thickness distribution shown in this article's section 3.3:*

*(1) In section 4.1 (see Fig. 13b for interpretation) we observe late R-shears offsetting older D- and R'-shears on the footwall side, terminating in the clay smear, and closely associated with the characteristic stair-steps.*

*(2) In section 5 (Fig. 14) we observe late formed R'-shears causing stair-steps in clay smear. These R'-shears with 1-2 cm displacement offset earlier formed D-shears. While growing across the footwall they cut through the undeformed source clay beds which provide weak slip zones and allows the R'-shears to be almost horizontal. Finally, the R'-shears protrude towards the main clay smear offsetting the clay-smear/sand contact in the footwall and hence forming triangular stair-steps.*

*(3) In dynamic observations of clay smear formation in sandbox models (cf. analogue models of Noorsalehi-Garakani et al., 2013; Schmatz et al., 2010b, a) steps in clay smear were observed to form without the presence of R- or R'-shears as a result of fault segmentation and erosion of clay lenses.*

*The model explaining observation (1) involves a highly strained clay smear with fully developed D- and R'-shears. In a later stage R shears on the footwall side continue to accommodate offset or nucleate, truncating the clay smear and the older D- and R'-shears (Fig. 15a). This is combined with a redistribution of the shearing clay to maintain continuity. Where R-shears truncate the clay smear it locally becomes very thin. With enough offset on the R-shears this process may be able to form holes in a clay smear (cf. paragraph 5.4).*

*A model explaining observation (2) is similar to the first model. In a first step faults develop clay smears and associated D-shears around the smears. Later R'-shears develop in the footwall or continue to accommodate strain while finally truncating the clay smear with low angle dips (Fig. 15b). As in the first model the clay in our outcrop has to be redistributed within the smear to maintain continuity. At locations where R'-shears truncate the clay smear it can get very thin and Kristensen et al. (2013) reported a clay smear that was disrupted by R'-shears with offsets larger than the clay smear thickness.*

*Finally, observation (3) can be explained by a model consisting of two processes. First, a clay lens is incorporated into the clay smear by fault segmentation (i.e. step-wise migration of the fault dip towards the footwall). At this point there are steps on both hanging- and footwall side off the clay smear. Secondly, continuing shear on the hanging-wall side of the smear then erodes clay at the hanging-wall side (Fig. 15c). This results in a straight surface on the hanging-wall side, while on*

*the footwall side a step remains visible. This process forms stair-stepping geometries without the presence of R- or R'-shears and without offsetting existing D-shears. No distinct thinning of the clay smear occurs."*

7. *Lines 482-483. In the model there is the assumption of circular clay fragments. Since clay minerals are platy minerals I suggest to better justify this assumption.*

→ **The circular clay fragments are not intended to resemble individual clay particles, but rather clay fragments in mm to cm scale as observed and described in e.g. Fig. 12. However, you are right; this is a simple assumption, which we think is ok for a first model with the sole aim of pointing out some relevant parameters that control grain-scale mixing.**